# A single transcription factor facilitates an insect host combating *Bacillus thuringiensis* infection while maintaining fitness

Zhaojiang Guo [1,2,6] ✉, Le Guo[1,6], Jianying Qin[1,6], Fan Ye[1,6], Dan Sun[1,6], Qingjun Wu[1], Shaoli Wang[1], Neil Crickmore [3], Xuguo Zhou [4], Alejandra Bravo[5], Mario Soberón[5] & Youjun Zhang [1] ✉

Maintaining fitness during pathogen infection is vital for host survival as an excessive response can be as detrimental as the infection itself. Fitness costs are frequently associated with insect hosts countering the toxic effect of the entomopathogenic bacterium *Bacillus thuringiensis* (Bt), which delay the evolution of resistance to this pathogen. The insect pest *Plutella xylostella* has evolved a mechanism to resist Bt toxins without incurring significant fitness costs. Here, we reveal that non-phosphorylated and phosphorylated forms of a MAPK-modulated transcription factor fushi tarazu factor 1 (FTZ-F1) can respectively orchestrate down-regulation of Bt Cry1Ac toxin receptors and up-regulation of non-receptor paralogs via two distinct binding sites, thereby presenting Bt toxin resistance without growth penalty. Our findings reveal how host organisms can co-opt a master molecular switch to overcome pathogen invasion with low cost, and contribute to understanding the underlying mechanism of growth-defense tradeoffs during host-pathogen interactions in *P. xylostella*.

Microbes, including bacteria, virus, and fungi, are the most abundant (over $10^{30}$) living beings on this planet[1]. While some microbes are beneficial, many others are pathogens, which colonize and shape the environmental adaptability of their host organisms[2]. Microbial pathogens and their hosts have developed a delicate and complex relationship during millions of years of co-evolution. During host–pathogen interactions, pathogens can deploy diverse virulence factors (toxins, effectors, etc.) as biochemical weapons to subdue their plant and animal hosts, while host organisms have also evolved sophisticated defense strategies to counter pathogen infection[3–5]. Nonetheless, enhanced immune defenses to pathogens in plant and animal hosts frequently compromise growth, development, and reproduction, leading to the need for growth-defense tradeoffs.

Coordinating growth-defense tradeoffs to minimize fitness costs when coping with pathogens is vital for the well-being and survival of plant and animal hosts. In recent decades, tremendous advances have been made in dissecting the role of gene regulation, signaling pathways, and immunity networks underlying growth and defense, which are poised to deliver a greatly improved understanding of tradeoff tactics[6–16].

*Bacillus thuringiensis* (Bt) is a gram-positive entomopathogenic bacterium that can employ diverse insecticidal toxins (e.g., Cry and Vip proteins) to facilitate infection of its insect hosts[17,18]. These toxins create pores in the gut endothelium, which facilitate entry of the bacterial cell. The binding of Bt toxins to specific receptors on the surface of the midgut epithelium is a crucial step for exerting virulence, and the primary midgut receptors include cadherin (CAD),

[1]Department of Plant Protection, Institute of Vegetables and Flowers, Chinese Academy of Agricultural Sciences, Beijing 100081, China. [2]Guangdong Laboratory for Lingnan Modern Agriculture, Guangzhou 510642, China. [3]School of Life Sciences, University of Sussex, Brighton BN1 9QE, UK. [4]Department of Entomology, University of Kentucky, Lexington, KY 40546-0091, USA. [5]Departamento de Microbiología Molecular, Instituto de Biotecnología, Universidad Nacional Autónoma de México, Cuernavaca 62250, México. [6]These authors contributed equally: Zhaojiang Guo, Le Guo, Jianying Qin, Fan Ye, Dan Sun. ✉e-mail: guozhaojiang@caas.cn; zhangyoujun@caas.cn

alkaline phosphatase (ALP), aminopeptidase N (APN), and ATP-binding cassette (ABC) transporters[19,20]. The diamondback moth, *Plutella xylostella* (L.), a most destructive and globally distributed agricultural pest, was the first insect documented as developing high-level resistance to Bt sprays in the field[21]. As with resistance to other xenobiotics, insect resistance to Bt toxins is generally accompanied by fitness costs (growth retardation, low survival rate, decreased fecundity, etc.)[22–24]. However, no obvious fitness costs have been observed in many resistant *P. xylostella* strains[22,25], rendering it an excellent model insect to study the underlying molecular mechanisms of how insect hosts can efficiently overcome Bt toxicity. Recently, we have established that altered expression of various midgut-expressed genes, *trans*-regulated by a hormone-activated mitogen-activated protein kinase (MAPK) signaling pathway, is linked to Bt resistance in different *P. xylostella* strains[14,26,27]. In these strains, genes encoding proteins that can act as Bt toxin receptors (ABCB1, ABCC2, ABCC3, ABCG1, ALP, APN1, and APN3a) are downregulated, whereas non-receptor paralogs (ABCC1, APN5, and APN6) are up-regulated and are believed to compensate physiologically for the loss of the proteins acting as midgut receptors. The precise mechanism by which the differential expression of these midgut proteins is achieved has remained unresolved.

Here, we reveal the regulatory framework adopted by the host insect to tackle the effect of Bt toxins. The transcription factor (TF) fushi tarazu factor 1 (FTZ-F1) promotes the expression of multiple receptor-encoding genes, whereas a phosphorylated form, activated by the MAPK signaling pathway, promotes the expression of the non-receptor paralogs. The MAPK-induced phosphorylation of FTZ-F1 reduces the cellular pool of non-phosphorylated TF, thus simultaneously reducing the expression of the receptors and increasing the expression of the non-receptor paralogs. This elegant strategy uses a single, pivotal, TF to protect the host insect from the effect of the pathogen whilst maintaining physiological fitness.

## Results

### FTZ-F1 activates the expression of diverse midgut genes

MAPK cascades typically activate downstream TFs via phosphorylation in order to control gene transcription[28]. Our recent quantitative phosphoproteomics data[27] showed that three TF proteins: nuclear receptor fushi tarazu factor 1 (FTZ-F1), prolactin regulatory element-binding protein (PREB), and RB1-inducible coiled-coil protein 1 (RB1CC1), displayed differential phosphorylation levels in the midgut of a susceptible strain (DBM1Ac-S) of *P. xylostella* compared to a near-isogenic resistant variant (NIL-R). To explore whether these TFs could control the expression of midgut Bt toxin receptor and non-receptor paralogous genes, the coding sequences of the TFs and the promoter sequences of the differentially expressed midgut genes (Supplementary Note 1) were cloned. All promoter sequences were mapped from the translation start site (TSS) since this provided a fixed reference point which, by default, is downstream of the promoter. The *FTZ-F1* gene encodes two protein isoforms, including αFTZ-F1 and βFTZ-F1, which contain a unique N-terminus followed by a conserved C-terminus (Supplementary Fig. 1). Aside from the different N-terminal sequence, the two protein isoforms of FTZ-F1 in *P. xylostella* contain the same DNA-binding domain (DBD) and ligand-binding domain (LBD), both of which have identical coding sequences in the susceptible and resistant strains (Supplementary Fig. 1). A dual-luciferase reporter assay was carried out to assess the regulatory effect of these factors on the expression of the midgut genes. Both αFTZ-F1 and βFTZ-F1 significantly increased the promoter activity of a number of receptor-encoding genes (*APN1*, *APN3a*, *ABCC2*, *ABCC3*, and *ABCG1*) and their non-receptor paralogous genes (*APN5*, *APN6*, and *ABCC1*), while the expression of two other receptor-encoding genes (*ALP* and *ABCB1*) was not affected by these TFs. In contrast, neither PREB nor RB1CC1 had any effect on any of these genes (Fig. 1a). Although the fold changes observed in the presence of co-transfected FTZ-F1 were

modest, they are consistent with other studies involving this TF[29]. The spatial and temporal transcription profiles of *αFTZ-F1* and *βFTZ-F1* were determined and showed that both isoforms were expressed in all tissues studied (Supplementary Fig. 2). Both forms peaked during development with *αFTZ-F1* showing high expression in the female adult and *βFTZ-F1* being induced at the onset of molting and pupation (Supplementary Fig. 2). This temporal variation of *βFTZ-F1* is consistent with previous reports showing the role of this factor in insect development and metamorphosis[30,31]. The expression atlas of FTZ-F1 implies that it is likely to be involved in controlling different physiological processes in *P. xylostella*, and perhaps one or more of these are highly regulated in female adults. The insect's response to Bt toxins described here is restricted to the midgut epithelial tissues, and so we presume that there is some mechanism, perhaps involving scaffolds, to target this FTZ-F1/MAPK response to this tissue. Considering that the two isoforms of FTZ-F1 contained the same DBD and LBD, and showed a similar regulatory effect, we just used αFTZ-F1 for subsequent analyses.

### Identification of functional FBSs in midgut gene promoters

To support the concept that FTZ-F1 could regulate the midgut genes, we searched for FTZ-F1 binding sites (FBSs) in the promoter regions of these genes using bioinformatic analyses. Putative FBSs were found associated with all these midgut genes, including those whose expression was not affected by FTZ-F1 (Supplementary Fig. 3 and Supplementary Note 1). In order to further identify the functional FBSs in these midgut genes, we utilized a reporter assay combined with gene promoter truncations. By creating a series of gene deletions, removing one FBS at a time, in each of the FTZ-F1 responsive midgut genes, we were able to identify functional sites by observing at which point the enhancement effect of FTZ-F1 was lost. Functional FBSs were preliminarily identified by this approach for all of the receptor-encoding genes: FBS2 between −500 and −469 in the *APN1* promoter (Fig. 1b), FBS4 between −1130 and −1091 in the *APN3a* promoter (Fig. 1c), FBS5 between −100 to −70 in the *ABCC2* promoter (Fig. 1d), FBS5 between −150 and −130 in the *ABCC3* promoter (Fig. 1e) and FBS4 between −222 and −152 in the *ABCG1* promoter (Fig. 1f). All these FBSs were similar to the canonical FTZ-F1 binding motif (5′-YCAAGGYCR-3′) found in mammals and *Drosophila*[32,33]. Intriguingly, there was no correlation between any putative FBS and expression for any of the non-receptor genes, despite their expression being controlled by FTZ-F1. For *ABCC1*, it appeared that FTZ-F1 interacted with a region between −1095 and −785 since the enhancement effect was lost when this region was deleted, however, no FBS had been identified in this region (Fig. 1g). For *APN5* and *APN6*, the deletion of regions upstream of −1100 and downstream of −380 respectively resulted in the loss of enhancement and also lacked any FBSs (Fig. 1h, i). To hone in further, two more sets of deletions were made in these regions. In the first set of deletions, the putative binding sites were narrowed to a sequence of 100 bp or less. In the second set of deletions, these sub-regions were split into 3 or 4 to further narrow down the regions to less than 30-bp sections [APN5-P(−1200/−1170), APN6-P(−266/−244), and ABCC1-P(−839/−823)] that we hypothesized could contain the potential binding sites for FTZ-F1 (Fig. 2a–c). TF binding sites (TFBSs) are usually small (about 6–12 bases) and can vary in sequence, which can make identification difficult[34]. To functionally pinpoint the binding sites in these non-receptor genes, we constructed reporter plasmids with 5–6 nucleotide mutations within the identified regions and examined their responses to FTZ-F1 in reporter assays. Mutations in M4 and M5 reduced the effect of FTZ-F1 on the *APN5* gene (Fig. 2d). Mutations in M2 and M3 lead to reduced FTZ-F1 induced promoter activity of *APN6* (Fig. 2e). For *ABCC1*, the effect was observed after mutations in M3 and M4 (Fig. 2f). Thus, the functional FBSs in non-receptor genes were associated with APN5-P(5′-TAAAGTCGGTTT-3′), APN6-P(5′-CATACAGTCTT-3′), and ABCC1-P(5′-GTACAGTCAC-3′). Based on these results, a putative FBS was identified as 5′-TA(A/C)AGTC-3′.

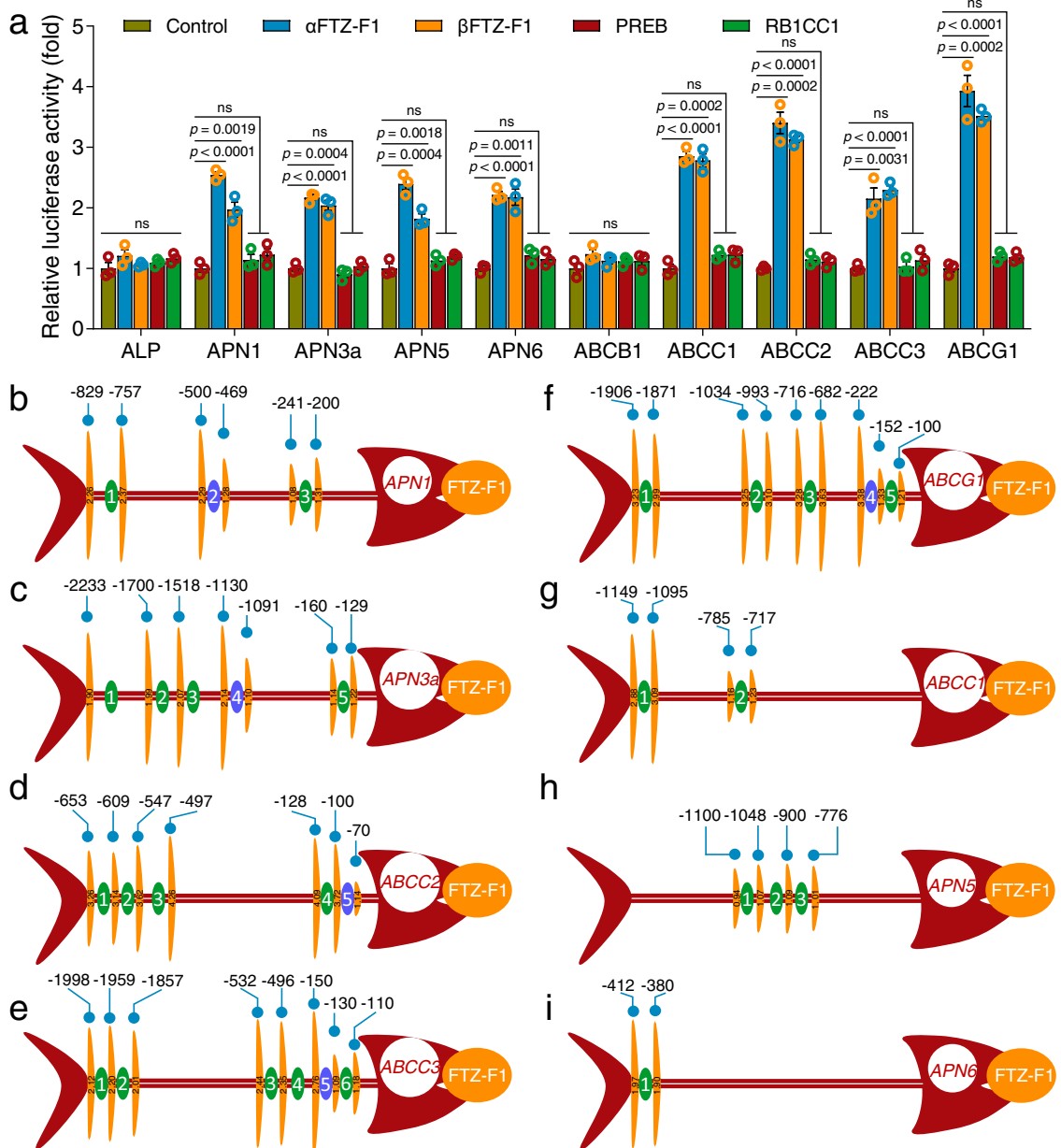

**Fig. 1 | FTZ-F1 regulates the expression of multiple midgut Cry toxin receptors and non-receptor paralogous genes. a** Effects of four TFs on the promoter activity of Bt receptor genes and non-receptor paralogous genes. Each pAc5.1-TF expression vector was co-transfected with a pGL4.10-promoter reporter plasmid into S2 cells to detect luciferase activity. An empty pAc5.1 vector was used as a control. The relative luciferase activity (fold) was calculated based on the value of the control, which was assigned an arbitrary value of 1. Differences between control and TF-treated groups were tested by one-way ANOVA with Tukey's test. Data were presented as mean values ± SEM ($n = 3$), ns, not significant, $p$ values are shown. **b–i** Preliminary identification of functional FBSs in the promoters of midgut genes by dual-luciferase reporter assays. The FTZ-F1 expression vector was co-transfected with various truncated constructs of midgut gene promoters to identify functional FBSs. The empty pAc5.1 vector was used as a control. The data of relative luciferase activity (fold) represent the mean value and was calculated based on the value of the control ($n = 3$), which was assigned an arbitrary value of 1. The results are presented as fish-like shapes. The head shows the target gene, and the orange ellipse by the mouth denotes the TF FTZ-F1. The horizontal red fishbone represents the promoter region, and the numbered ellipses represent the predicted FBSs, where present the purple ellipse represents the potential functional FBS. The height of the vertical orange fishbone represents the relative luciferase activity (fold) of the different truncations of a given promoter (the specific values are represented by vertical black Arabic numerals). The horizontal numbers represent the nucleotide position of the different truncations relative to the start codon. Source data are provided as a Source Data file.

## Different forms of FTZ-F1 bind to distinct DNA motifs

Preliminary identification of the functional FBSs indicated that FTZ-F1 appeared to regulate receptor genes and non-receptor paralogous genes by binding to distinct motifs. Given that in our phosphoproteomic analysis, we observed that FTZ-F1 phosphorylation was increased in the resistant strain[27], we considered the hypothesis that phosphorylation influences the regulatory role of FTZ-F1. To confirm phosphorylation, an EGFP-FTZ-F1 fusion protein was heterologously expressed in Sf9 cells and immunoprecipitated in order to perform mass spectroscopy (MS) (Fig. 3a). The resulting MS data indicated the possibility of four phosphorylation sites of threonine (T): T288 (Fig. 3b), T361, T538 and T544 (Supplementary Fig. 4 and Supplementary Data 1). We predicted the phosphorylation of FTZ-F1 computationally by disorder-enhanced phosphorylation predictor (DEPP), which highlighted T288 as having the highest score (Fig. 3c). In general, an alanine (A) substitution at the phosphorylation site can mimic the

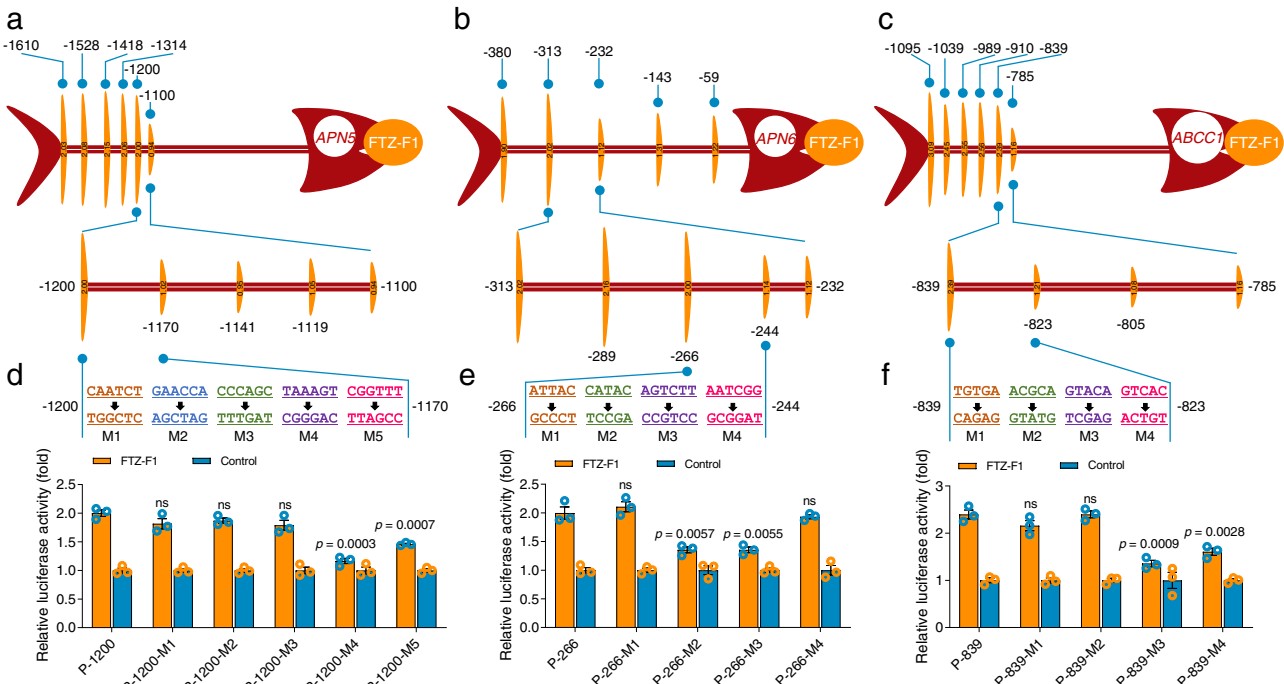

**Fig. 2 | Preliminary identification of the functional binding sites in the promoters of the non-receptor genes *APN5*, *APN6*, and *ABCC1* by a dual-luciferase reporter assay.** The FTZ-F1 expression vector was co-transfected with various truncated promoters of non-receptor genes *APN5* (**a**), *APN6* (**b**), and *ABCC1* (**c**). The results are presented using the same fish-like structure as Fig. 1. The lower "bones" represent the second set of deletions created within the region identified as containing the functional binding site from the initial set of deletions. The horizontal numbers represent the nucleotide position of the different truncations relative to the start codon. **d–f** The effect of FTZ-F1 on wild-type or mutated promoters of *APN5*, *APN6*, and *ABCC1* genes. A series of recombinants comprising 5–6 base mutations in each of the promoter regions was constructed and co-transfected with an FTZ-F1 vector to precisely identify the position of the functional FBSs. An empty pAc5.1 vector was used as a control (**a–f**). The relative luciferase activity (fold) was calculated based on the value of the control, which was assigned an arbitrary value of 1. Data were presented as mean values (**a–c**) and mean values ± SEM (*n* = 3) (**d–f**), ns, not significant, *p* values are shown. Differences between wild-type and mutated promoters were tested by one-way ANOVA with Tukey's test (**d–f**). Source data are provided as a Source Data file.

non-phosphorylated form of a protein, whereas an aspartic acid (D) can imitate the phosphorylated form[11]. We therefore created various mutated FTZ-F1 proteins, FTZ-F1$^{T288A(-P)}$, FTZ-F1$^{T361A(-P)}$, FTZ-F1$^{T538A(-P)}$, and FTZ-F1$^{T544A(-P)}$ to abolish phosphorylation capacity, and FTZ-F1$^{T288D(P)}$, FTZ-F1$^{T361D(P)}$, FTZ-F1$^{T538D(P)}$, and FTZ-F1$^{T544D(P)}$ to mimic phosphorylation. In reporter gene assays, FTZ-F1$^{T288A(-P)}$ activated the receptor genes but did not affect the non-receptor ones. In contrast, FTZ-F1$^{T288D(P)}$ induced the non-receptor genes but had little effect on the receptor gene promoters (Fig. 3d). However, neither form of the T361, T538, and T544 FTZ-F1 mutant proteins showed any significant regulatory activity changes to these midgut genes (Supplementary Fig. 5). Although four sites on FTZ-F1 were found to be phosphorylated in vivo, not all four post-translational modifications are necessarily important. The results indicated T288 as the most likely functional phosphorylation site and the other three could simply represent phosphorylation events that have no biological significance, or are not involved in the function of FTZ-F1 in this context.

To further investigate whether this occurred through binding to different DNA motifs (Fig. 3e, f), FTZ-F1$^{T288A(-P)}$ and FTZ-F1$^{T288D(P)}$ were co-transfected with promoter constructs containing the wild-type or mutated binding sites that we had previously identified as being the functional FBSs (Figs. 1, 2). Our data showed that the mutated binding sites in the receptor genes blocked the transcription enhancing effect of non-phosphorylated FTZ-F1$^{T288A(-P)}$ (Fig. 3g). For the non-receptor genes, mutation of the variant FBS motif blocked the enhancing effect of the phosphorylated mimic FTZ-F1$^{T288D(P)}$ (Fig. 3h). The form of FTZ-F1 expressed in the co-transfection experiments (Fig. 1a) could activate both receptor genes and non-receptor paralogous genes, implying that it might be partially phosphorylated in the S2 cell system. To test this possibility, nucleoproteins of S2 cells with and

without recombinant FTZ-F1 expression were collected, phosphorylated FTZ-F1 was separated from non-phosphorylated FTZ-F1 in a Phos-tag SDS-PAGE gel, and both forms detected using an FTZ-F1 polyclonal antibody raised against our recombinant protein. Both phosphorylated and non-phosphorylated FTZ-F1 proteins were only detected in the cells expressing the FTZ-F1 protein (Supplementary Fig. 6). These data supported the hypothesis that non-phosphorylated FTZ-F1 binds to the canonical FBS motif, whereas the phosphorylated form binds to the variant motif (5′-TA(A/C)AGTC-3′) hereafter named as FBS$^P$.

An electrophoretic mobility shift assay (EMSA) and a yeast one-hybrid assay (Y1H) were then conducted to further confirm the direct binding of the different forms of FTZ-F1 to these two motifs. In the EMSA assay, the non-phosphorylated FTZ-F1$^{T288A(-P)}$, but not the phosphorylated FTZ-F1$^{T288D(P)}$, specifically bound to the canonical FBS probe (Fig. 4a). Phosphorylated FTZ-F1$^{T288D(P)}$, but not the non-phosphorylated form, showed specific binding to the FBS$^P$ probe (Fig. 4b). In the Y1H assays, the yeast strains that were co-transformed with FTZ-F1$^{T288A(-P)}$ and the canonical FBS or with FTZ-F1$^{T288D(P)}$ and FBS$^P$ grew normally in the selective medium, whereas strains containing the prey proteins and mutated motifs did not grow (Fig. 4c). These studies further supported the hypothesis that non-phosphorylated FTZ-F1 activates midgut receptor genes via the FBS, while phosphorylated FTZ-F1 regulates midgut non-receptor genes via FBS$^P$. Since altering the phosphorylation status of other TFs has previously been shown to relocate the protein out of the nucleus[35], a subcellular localization study was performed and confirmed that both the non-phosphorylated and phosphorylated FTZ-F1 proteins are located in the nucleus—the site at which they would be expected to act as TFs (Fig. 4d).

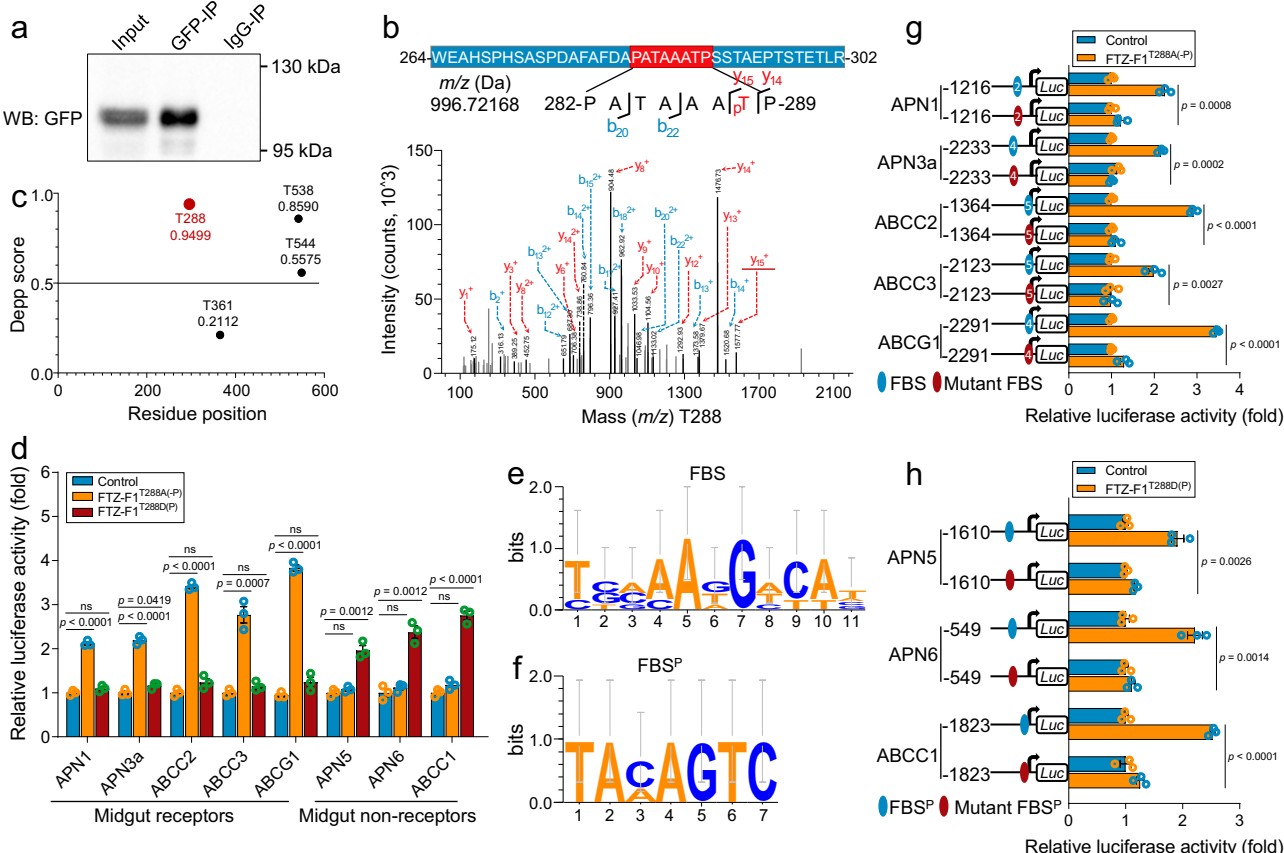

**Fig. 3 | Identification of the functional phosphorylation sites for FTZ-F1.**
**a** Immunoprecipitation of exogenous FTZ-F1 from Sf9 cells using anti-GFP. **b** The putative phosphorylation site T288 of FTZ-F1 was identified by LC-MS/MS. The precursor ion, which ranges from 264 to 302, is shown in the top band with the amino acids in white letters, and the representative fragment ion is shown with a red background with detailed information. The identified phosphorylation site T288 is represented by a red letter and subscripted p, while the phosphorylated peptide on the spectrum is underlined in red. **c** The DEPP score of the four putative phosphorylation sites was computed using the DEPP software in silico. **d** The regulatory effect of phosphorylated FTZ-F1$^{T288D(P)}$ and non-phosphorylated FTZ-F1$^{T288A(-P)}$ on midgut genes. A substitutes T mimicking dephosphorylation and D

substitutes T mimicking sustained phosphorylation. **e, f** WebLogo plots highlight the potential functional FBS in receptor gene promoters (**e**) and FBS$^P$ in non-receptor gene promoters (**f**). **g** Effect of non-phosphorylated FTZ-F1$^{T288A(-P)}$ on the activity of receptor promoters with either a wild-type or a mutant FBS. For clarity, only the functional FBSs are shown. **h** Effect of phosphorylated FTZ-F1$^{T288D(P)}$ on the activity of non-receptor promoters with either a wild-type or a mutant FBS$^P$. The empty pAc5.1 vector was used as a control (**d, g, h**), and the relative luciferase activity (fold) was calculated based on the value of the control, which was assigned an arbitrary value of 1. Data were presented as mean values ± SEM ($n = 3$), ns, not significant, $p$ values are shown. One-way ANOVA with Tukey's test was used for comparison. Source data are provided as a Source Data file.

## FTZ-F1 phosphorylation is associated with Cry1Ac resistance

To verify whether FTZ-F1's hypothesized mode of action associates with the Cry1Ac resistance phenotype in *P. xylostella*, we detected the transcript and protein levels of FTZ-F1 in the midgut tissue of different Cry1Ac-susceptible and resistant larvae. The data showed similar mRNA and protein levels of FTZ-F1 among the different strains (Fig. 5a, b), while the level of phosphorylated FTZ-F1 was observed to be higher in all four resistant strains compared to the susceptible DBM1Ac-S strain (Fig. 5b). These data were consistent with the hypothesis that the differential phosphorylation of FTZ-F1 in vivo might be associated with Cry1Ac resistance in *P. xylostella*.

To establish whether FTZ-F1 does actually modulate midgut gene expression in vivo, an RNAi assay was carried out. Silencing of *FTZ-F1* in larvae of the resistant strain NIL-R was accompanied by reduction in both phosphorylated and non-phosphorylated FTZ-F1 (Fig. 5c), as well as a decrease in the transcripts of all the midgut genes except *ABCB1* and *ALP* (Fig. 5d). Additionally, the susceptibility of *FTZ-F1*-silenced NIL-R larvae to Cry1Ac was significantly decreased compared to the untreated controls (Fig. 5e). When similar experiments were performed with the susceptible strain DBM1Ac-S, *FTZ-F1* silencing was again accompanied by a decrease in FTZ-F1 protein (Fig. 5f) and downregulation of receptor gene expression (Fig. 5g). Since very little

phosphorylated FTZ-F1 was naturally present in the susceptible strain, the reduction in this form was less significant as a result of RNAi (Fig. 5f) and this would explain the negligible effect of RNAi on non-receptor gene expression (Fig. 5g). The downregulation of receptor genes in DBM1Ac-S associated well with susceptibility to toxin as dsFTZ-F1-treated larvae presented a significant reduction in Cry1Ac-induced mortality (Fig. 5h).

## The MAPK cascade regulates the phosphorylation of FTZ-F1

We have previously shown that the activated MAPK signaling pathway can induce Bt resistance without significant fitness costs in *P. xylostella* and is initiated by increased expression of *MAP4K4*[14,26,27]. Generally, TFs downstream of the MAPK cascade are responsible for transmitting the signal to functional genes[28]. Moreover, the identified functional phosphorylation site T288 in FTZ-F1 is within a MAPK consensus target sequence (Supplementary Fig. 7)[36]. To explore whether the transcriptional activity of FTZ-F1 is controlled by MAPK-mediated phosphorylation of FTZ-F1, we investigated the effect of *MAP4K4* silencing on FTZ-F1 expression and phosphorylation. Silencing of *MAP4K4* expression in the resistant strain NIL-R did not change the transcript or protein levels of FTZ-F1 (Fig. 6a, b), but significantly reduced its phosphorylation level (Fig. 6b).

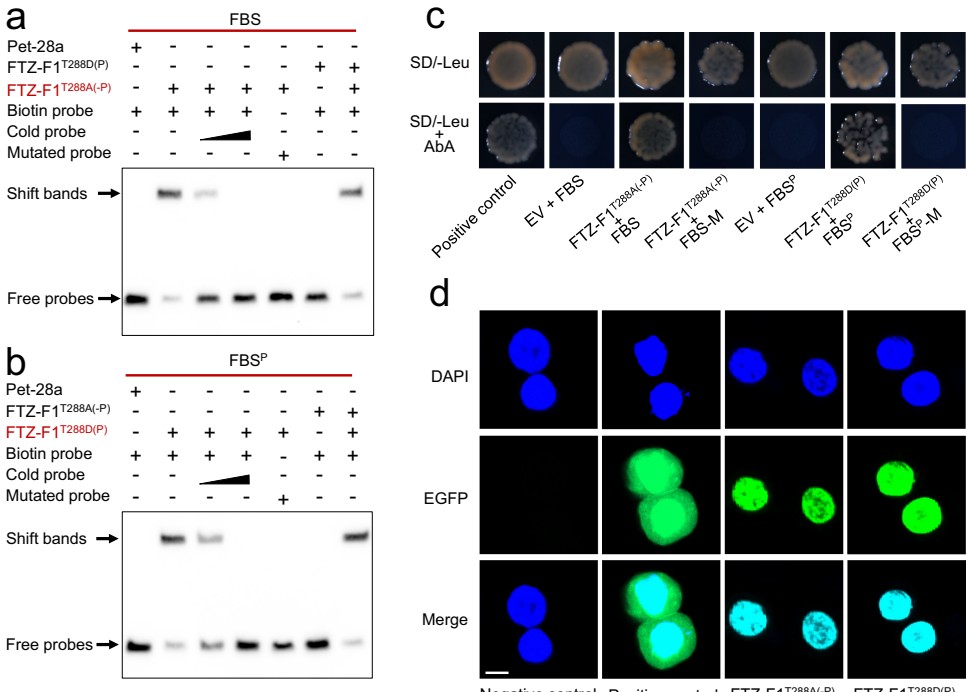

**Fig. 4 | Phosphorylated and non-phosphorylated FTZ-F1 preferentially bind to distinct DNA motifs and both function in the nucleus. a, b** EMSA assay validates that the non-phosphorylated FTZ-F1$^{T288A(-P)}$ (**a**) and the phosphorylated FTZ-F1$^{T288D(P)}$ (**b**) specifically bind to the FBS (**a**), and FBS$^P$ (**b**) respectively. The concentrations of the wild-type and mutant probes were 20 fmol; the concentrations of competing cold probes were 100 and 500 fmol. The mutant sequences were 5′-CGCACA-CACGT−3′ for FBS and 5′-GGCTCCGAAC-3′ for FBS$^P$. **c** Y1H assays verifying the direct binding of non-phosphorylated FTZ-F1$^{T288A(-P)}$ to FBS, and the direct binding of phosphorylated FTZ-F1$^{T288D(P)}$ to the FBS$^P$ using wild-type or mutated binding sites as described below. EV empty vector; positive control, pGADT7-p53 + pABAi-p53. **d** Subcellular localization of non-phosphorylated FTZ-F1$^{T288A(-P)}$ protein and the phosphorylated FTZ-F1$^{T288D(P)}$ protein. The nuclei were stained with DAPI, the Sf9 cells or cells transfected with the empty plasmids (Pie-EGFP-N1) were used as controls. Scale bar: 10 µm. Source data are provided as a Source Data file.

MAPK cascades typically regulate downstream TFs through the p38, JNK, or ERK pathways[28]. To probe the pathway responsible for phosphorylating FTZ-F1, resistant NIL-R larvae were fed specific inhibitors of p38, ERK, and JNK, respectively. As with *MAP4K4* silencing, the inhibitor treatments had little effect on the mRNA or protein levels of FTZ-F1 (Fig. 6d, e). Compared to the control, however, phosphorylated FTZ-F1 was decreased in larvae treated with the p38 inhibitor, with possible downregulation also seen in larvae treated with ERK or JNK inhibitors (Fig. 6e). Moreover, silencing of *MAP4K4* or inhibitor treatment significantly recovered larval susceptibility in the resistant NIL-R strain (Fig. 6c, f).

**The importance of non-receptor genes for maintaining fitness**
FTZ-F1 can modulate both receptor and non-receptor expression, the role of non-receptor genes, however, has not yet been experimentally tested, thus, we wanted to directly test the hypothesis that expression of the non-receptor paralogs is important for maintaining fitness. To ascertain the contribution of non-receptor genes, a series of homozygous mutant strains were generated using CRISPR/Cas9 (Supplementary Fig. 8).

A CRISPR/Cas9-induced knockout of the *ABCC1* gene was performed ab initio to introduce a 2-bp deletion in the *ABCC1* locus in the *P. xylostella* NIL-R resistant strain (Supplementary Fig. 9c, d). A non-destructive method involving direct sequencing and TA cloning was used to screen mutant individuals, and a germline transformation strategy was used to construct a stable homozygous mutant strain (C1KO) (Fig. 7a top, Supplementary Fig. 8a, and Supplementary Table 1). Next, based on the adjacent location of *APN5* and *APN6* genes in the *P. xylostella* genome, a dual sgRNA CRISPR/Cas9 method was used to simultaneously knock out both of these genes from the NIL-R

strain (Supplementary Fig. 9a and Supplementary Table 2). PCR amplification using four gene-specific primers spanning *APN5* and *APN6* indicated successful mutagenesis (Supplementary Fig. 9b). A homozygous double-mutant strain (N6-5KO) with an approximately 13-kb deletion (Fig. 7a middle and Supplementary Fig. 8b) was created. Finally, we generated a homozygous *ABCC1/APN6/APN5* triple knock-out strain (C1/N6/N5KO) by introducing a mutation (5-bp deletion) in the *ABCC1* gene to the aforementioned N6-5KO strain (Fig. 7a bottom, Supplementary Fig. 8c, and Supplementary Table 1). Bioassays were subsequently conducted to detect any susceptibility differences to Bt Cry1Ac protoxin between the newly established strains along with DBM1Ac-S and NIL-R strains as control (Supplementary Table 3). These results showed that there were no significant differences in resistance level between the newly-built mutant strains (C1KO: 5108-fold, N6-5KO: 5034-fold, and C1/N6/N5KO: 4967-fold) and the parental resistant strain (NIL-R: 5169-fold) compared to the susceptible strain.

We had previously demonstrated that Cry1Ac was unable to bind to the non-receptor paralogs APN5 and APN6 (but could to APN1 or APN3a) when these proteins were ectopically expressed in Sf9 cells[14]. We now also show that in contrast to ABCC2 or ABCC3, Cry1Ac cannot bind to, nor affect the susceptibility of, Sf9 cells ectopically expressing ABCC1 (Supplementary Fig. 10). These data indicated that ABCC1, APN5, and APN6 play little or no role in determining the level of resistance. To determine whether the non-receptor paralogs compensate physiologically for the loss of receptors, a series of life-history traits were measured in the resistant strain (in which the receptor proteins are constitutively downregulated) and in the mutants where the non-receptor paralogs had been knocked out. Pupal morphology, pupation percentage, pupal weight, pupal duration, and hatching rate were assessed, and we observed that the single C1KO and double N6-

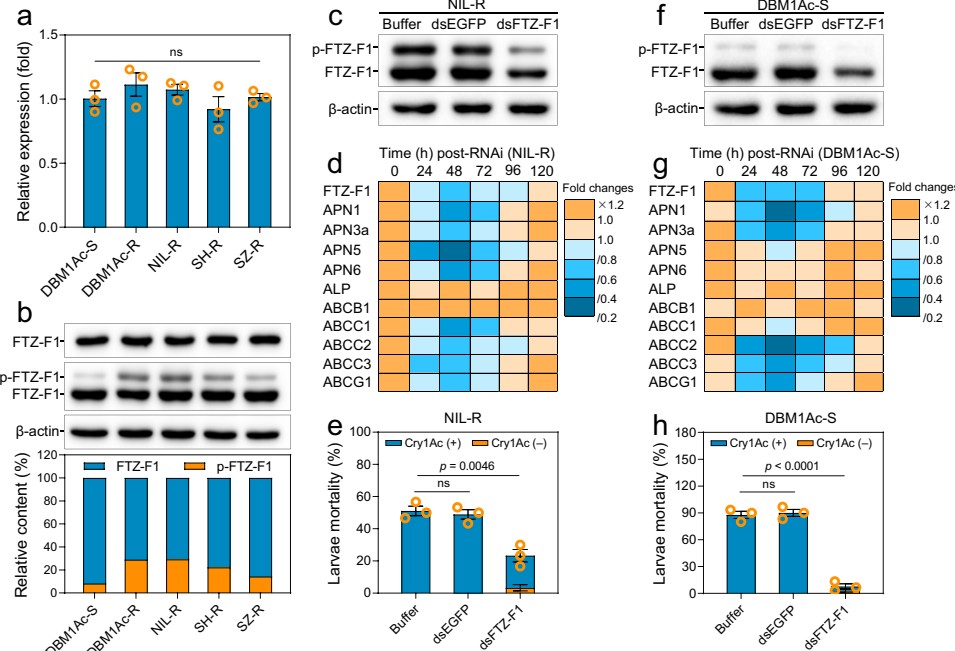

**Fig. 5 | Elevated phosphorylation of FTZ-F1 in vivo enhances resistance of *P. xylostella* larvae to Cry1Ac toxin. a** *FTZ-F1* transcript level in the larval midgut of a susceptible DBM1Ac-S strain and four resistant *P. xylostella* strains. The relative expression level was quantitated and normalized to the expression level of the *RPL32* gene and the value in the DBM1Ac-S strain was set as 1. **b** Protein expression and phosphorylation levels of FTZ-F1 in the larval midgut of the same five *P. xylostella* strains. Phosphorylated and non-phosphorylated FTZ-F1 proteins were separated on a Phos-tag SDS-PAGE gel, detected by anti-FTZ-F1, and quantitated by densitometry using the ImageJ 1.51 software and normalized to the β-actin.

**c–e** Effect of *FTZ-F1* silencing on phosphorylation of FTZ-F1 in the midguts (**c**), the transcript level of midgut genes (**d**), and larval susceptibility to an LC$_{50}$ concentration of Cry1Ac (3980 mg/L) (**e**) in NIL-R strain. **f–h** Effect of *FTZ-F1* silencing on phosphorylation of FTZ-F1 in the midguts (**f**), the transcript level of midgut genes (**g**), and larval susceptibility to an LC$_{90}$ concentration of Cry1Ac (2 mg/L) (**h**) in DBM1Ac-S strain. Data were presented as mean values ($n = 3$) (**d**, **g**) and mean values ± SEM ($n = 3$) (**a**, **e**, **h**), ns, not significant, *p* values are shown. One-way ANOVA with Tukey's test was used in **a**, **e**, **h** for comparison. Source data are provided as a Source Data file.

5KO mutant strains had significant differences when compared to both the susceptible DBM1Ac-S and resistant NIL-R strains. The differences were even more pronounced in the triple knockout strain C1/N6/N5KO, which showed extremely significant fitness costs in all of the tested parameters (Fig. 7b–f). These results indicated that the increased expression of non-receptor paralogs is responsible for diminishing the fitness costs of Cry1Ac resistance.

## Discussion

Insects are constantly jeopardized by pathogens in their natural habitats and thus must hold an efficient immunity weapon on the battlefield of pathogen invasion. An ideal evolutionary model for the host is to employ a key gene(s) to balance growth and defense with high efficacy and low cost. In this study, we uncovered a single TF (FTZ-F1) as a key modulator of a response allowing the host insect *P. xylostella* to defend against the invasion of Bt pathogens without significant fitness costs (Fig. 8). Phosphorylation of FTZ-F1 reduces the cellular levels of non-phosphorylated FTZ-F1, which, since this form activates their expression, results in the downregulation of physiologically important proteins that Bt toxins used as receptors. To compensate for the loss of these proteins, the phosphorylated form of FTZ-F1 then induces the expression of non-receptor paralogs. This response can be considered similar in essence to the Chinese traditional martial art "Tai Chi Chuan", "four ounces can move 1000 pounds", which means "accomplish a great task with little effort by clever maneuvers".

We had previously speculated that the expression of the non-receptor paralogs could compensate physiologically for the loss of the midgut proteins acting as receptors for the Bt Cry1Ac toxin[14]. In that work, it was shown that treatment of susceptible larvae with the hormone 20-hydroxyecdysone (20E) resulted in the downregulation of both the receptor and non-receptor genes and that those insects also

exhibited significant fitness costs. In subsequent work, we together knocked out the four midgut receptors ABCC2, ABCC3, APN1, and APN3a in the susceptible strain[37], which resulted in a high level of resistance to Cry1Ac. The knockouts did not result in any increase in expression in the non-receptor paralogs ABCC1, APN5, or APN6 and the quadruple knockout strain showed significant fitness costs. While these two results provided indirect evidence that the non-receptor paralogs could compensate physiologically for the loss of the receptors in the susceptible strain, we provide direct evidence for the first time here by inducing fitness costs in the resistant strain by knocking out these genes.

In most cases, the host must reassign energy from growth to defense in response to stress factors in their environment, including pathogens or xenobiotics, provoking a fitness penalty[22–24]. Costs associated with resistance to pathogens have been corroborated in a wide array of hosts, a substantial numbers of studies have demonstrated that immune activation contributes to resistance against invading pathogens in plants while compromising yield[8,9]. Genetic resistance or tolerance to microbe transmission in mammals, like COVID-19 and SARS-CoV, rendered by innate or adaptive immune responses aggravates fitness costs in the form of extra energy demands, affiliated damage to host tissues and severe multiple organ dysfunction, even reproductive deficiency[38–40]. Additionally, the evolution of resistance to phages and parasites carries fitness costs for bacteria, such as sacrificing growth rate and reduced fecundity[41,42]. Host insects refractory to pathogens and parasitic infection can also have decreased fecundity or prolonged development time[43]. Immune defense is like a double-edged sword, with the ability to either support survival or cause autoimmune syndromes. Thus, hunting a dominant driver of adaptive evolution for the host is of preeminent importance. TFs are crucial for a host to acclimatize to adversity created by biotic and abiotic factors. TFs often act as sites of signal convergence and concomitantly, signal-regulated TFs

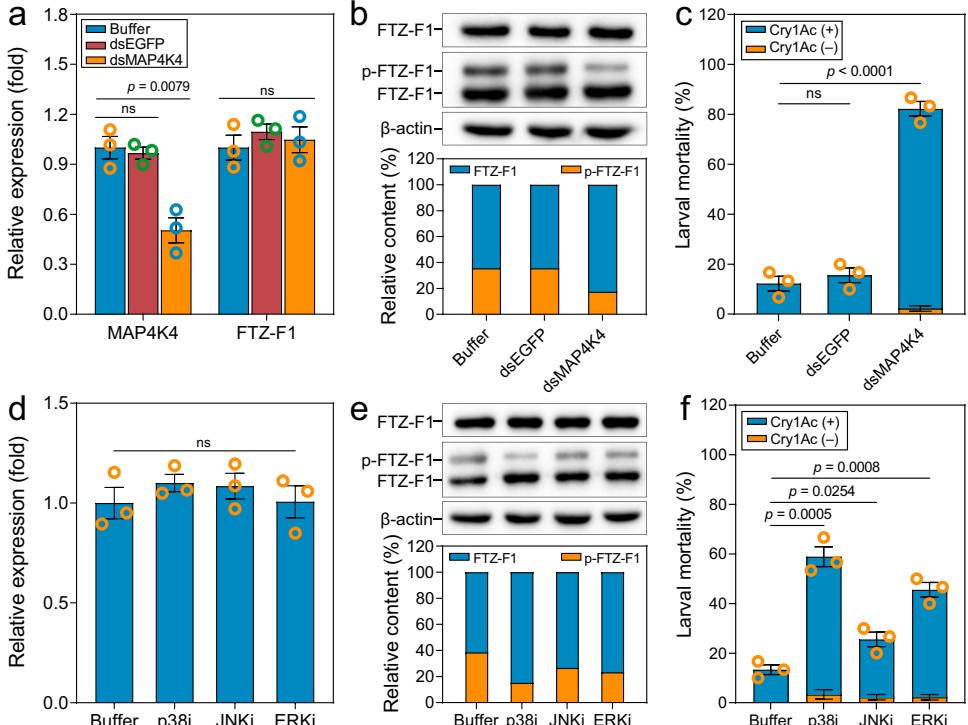

**Fig. 6 | MAPK cascades regulate the in vivo phosphorylation level of FTZ-F1.**
**a**–**c** Effect of silencing *MAP4K4* gene on FTZ-F1 mRNA expression (**a**), protein level and degree of phosphorylation (**b**), and larval mortality (**c**) in the resistant strain NIL-R. **d**–**f** Effect of specific inhibitors of p38, ERK, or JNK on FTZ-F1 mRNA expression (**d**), protein level and degree of phosphorylation (**e**), and larval mortality (**f**) in the resistant strain NIL-R. Relative mRNA expression levels (**a**, **d**) were quantitated and normalized to the expression level of the *RPL32* gene, the value for the

buffer treated strain was set as 1. Phosphorylated and non-phosphorylated FTZ-F1 proteins (**b**, **e**) were separated on a Phos-tag SDS-PAGE gel, detected by anti-FTZ-F1 and quantitated by densitometry using the ImageJ 1.51 software and normalized to the β-actin. Data were presented as mean values ± SEM (**a**, **c**, **d**, **f**). *n* = 3 biologically independent samples, ns, not significant, *p* values are shown. The differences between control and dsRNA-treated groups or inhibitor-treated groups were tested by one-way ANOVA with Tukey's test. Source data are provided as a Source Data file.

cooperate with other co-factors to establish transcription regulatory networks required for coordinating host stress responses with growth to maximize their fitness[44,45]. A single TF was found in rice to promote both yield and pathogen defense via regulating different target genes in disparate biological processes[11,13].

In insects, *FTZ-F1* was first identified in *Drosophila* three decades ago, encoding two protein isoforms: αFTZ-F1 and βFTZ-F1[46]. αFTZ-F1 is maternally supplied and acts as a cofactor of the homeodomain protein fushi tarazu (FTZ) to control embryonic pattern formation during early embryogenesis[47,48], while βFTZ-F1 is identified as a competence factor for stage-specific responses to ecdysone pulses and controls larval molting, metamorphosis, and pupal development[49,50]. Similar to the *Drosophila FTZ-F1* gene, we also identified two protein isoforms of the *FTZ-F1* gene in *P. xylostella*, which differ only in their N-terminal sequences. Although a role for FTZ-F1 in pathogen defense has not previously been described, a recent study has linked FTZ-F1 to resistance to a chemical insecticide in *P. xylostella*[51]. In response to pathogen attacks, hosts activate immune systems that are mediated through multifarious signaling pathways and hormone crosstalk, and it is an effective way to orchestrate physiological tradeoffs in a wide variety of organisms[14,52]. FTZ-F1 has been linked with stage-specific responses to ecdysone signaling[30,31], and recently, the power of 20E in resisting infection of Bt pathogens was consolidated in *P. xylostella*[14]. Combined with our previous observation, αFTZ-F1 and βFTZ-F1 display similar titer patterns to 20E during the feeding intermolt stage but differ at metamorphosis. Potentially, both forms of FTZ-F1 could promote the defense against Bt, but each is acting at different stages of development. The established roles of FTZ-F1 in insect growth and development provide a hint about how this versatile host TF has been recruited

into a pathogen response pathway that is linked to maintaining growth and development.

The precise determination of the binding site to the target protein is a prerequisite to deciphering the complex regulatory networks of TFs[53]. DNA binding motifs for many TFs in various species have been characterized during recent decades with the development of technologies such as ChIP-seq and ATAC-seq[54,55]. However, studies using these approaches mainly concentrate on TF binding to the primary target motifs and rarely characterize alternative ones[56]. Multiple studies have demonstrated that FTZ-F1 binds DNA with high affinity to 5′-YCAAGGYCR-3′, whereas little attention is known about functional non-canonical response elements. The aforementioned TF IPA1 in rice is an excellent example of where different forms regulate different genes. Phosphorylated IPA1 activates immune-related genes via a novel binding site distinct from that bound by non-phosphorylated IPA1[11], indicating that TFs with or without post-translational modifications, binding to different motifs, might be a more widespread phenomenon. Although the amino acid phosphorylated in FTZ-F1 does not form part of the conserved DNA binding motif, this does not preclude it from influencing binding as other studies have shown that this post-translational modification can alter the structure of distal parts of a DNA-binding protein[57]. TF-DNA binding specificity can be affected by additional layers of complexity, including interactions between TFs and other factors, TF changes (such as phosphorylation as found in this study), DNA motif context (including flanking sequences and DNA shape), and genomic features (such as chromatin accessibility and epigenetic information)[56,58]. The current atlas of binding motifs for FTZ-F1 and other TFs is still rather incomplete, identifying more targets and

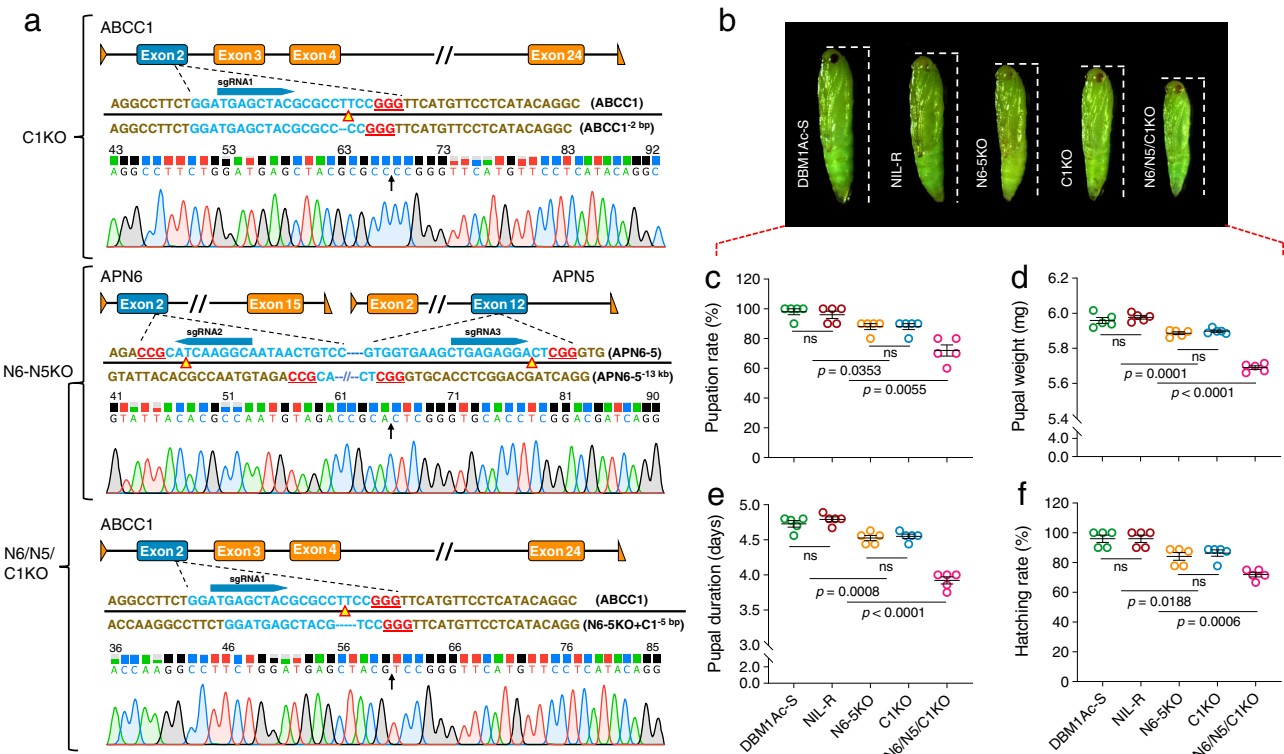

**Fig. 7 | CRISPR/Cas9-mediated knockout of non-receptor genes in *P. xylostella*.**
**a** Representative chromatogram of CRISPR/Cas9-induced mutation of *ABCC1* gene (top), double-gene knockout of *APN6* and *APN5* genes (middle), and triple mutant of *ABCC1/APN6/APN5* genes by introducing mutations on the base of the double-mutant (bottom). The cleavage site of Cas9 protein is indicated with a red-edged yellow triangle. The arrow indicates the junction position after deletion.
**b** Representative pupal morphology in *P. xylostella* susceptible DBM1Ac-S, resistant NIL-R, and non-receptor gene-edited strains. **c–f** Evaluation of fitness costs in non-

receptor gene mutant strains. A series of biological parameters were compared among non-receptor gene knockout strains (C1KO, N6-5KO, and C1/N6/N5KO) with the susceptible DBM1Ac-S and the parental NIL-R resistant strains. **c** Pupation rate. **d** Pupal weight. **e** Pupal duration. **f** Hatching rate. Data were presented as mean value ± SEM, *n* = 5 biologically independent samples with ten larvae per replicate, ns, not significant, *p* values are shown. One-way ANOVA with Tukey's test was used for comparison. Source data are provided as a Source Data file.

specific TFBSs remain a priority for decoding the complex regulatory action and new functions of TFs.

Based on the data gathered in this study, the MAPK-responsive FTZ-F1 is a key regulatory factor in the differential expression of receptors and non-receptor paralogs in *P. xylostella* (Fig. 8). Within the insect midgut, APN enzymes are primarily involved in the digestion process, and the fact that development-associated control of expression of different APNs has been observed suggests that different forms may have distinct roles in digestion[59]. We have also previously observed differences in the expression of various APNs between larval stages[14], suggesting that a mechanism for the differential expression of insecticidal APN genes exists. ABC transporter proteins have many transport and non-transport functions, although these remain unclear, particularly in arthropods[60]. The *P. xylostella* genes that we have demonstrated to be under the control of FTZ-F1 don't exactly match those that are differentially expressed in the resistant strain, upon Bt intoxication or following *MAP4K4* silencing[14]. In particular, no effect was seen on the two receptor-encoding genes *ALP* and *ABCB1*. Our recent studies have identified *cis*-acting mutations and *trans*-factors that regulate these two genes[61–64]. In both cases, *trans*-acting TFs were identified that were under the control of *MAP4K4*, as FTZ-F1 was. By acting via a partially independent pathway, allows some degree of protection against Bt if the primary FTZ-F1 pathway is not available for whatever reason.

Although the midgut protein-encoding genes investigated here are likely to be under normal homeostatic control, it is difficult to conceive a physiological process that would involve the pattern of differential expression observed in the resistant strain. The observed

process, however, is a complex, but elegant solution for overcoming the pathogenic effect of Bt toxins. The fact that the Bt Cry1Ac toxin has evolved to be able to target multiple proteins as receptors represent one side of the arms race that required the host to develop a sophisticated response mechanism. We conclude, therefore, that the pathogen response observed is not due to the co-option of an existing process but a specific mechanism that has arisen as a result of a long period of co-evolution between insects and Bt[65].

## Methods

### Insect strains and cell lines

Five *P. xylostella* strains, including one Bt-susceptible strain DBM1Ac-S and four Bt-resistant strains DBM1Ac-R, NIL-R, SZ-R, and SH-R were used in this study[26,66–68]. The susceptible DBM1Ac-S and field-evolved Bt-resistant DBM1Ac-R strains were provided by Drs. Jianzhou Zhao and Anthony (Tony) Shelton (Cornell University, USA) in 2003. Then, the near-isogenic Cry1Ac-resistant NIL-R strain was constructed in our laboratory in 2015 by six-time backcrossing between DBM1Ac-S and DBM1Ac-R along with Cry1Ac toxin selection. The lab-selected Cry1Ac-resistant SZ-R strain was collected in Shenzhen, China, in 2003 and generated by continuous selection with Cry1Ac protoxin. The lab-selected Bt-resistant SH-R strain was collected in Shanghai in 2005 and was treated with a Bt var. *kurstaki* (Btk) formulation (WP with a potency of 16,000 IU/mg, provided by Hubei Biopesticide Engineering Research Center, Hubei Academy of Agricultural Sciences, China). The DBM1Ac-R, NIL-R and SZ-R larvae present around 4500-, 5000-, and 500-fold resistance to Cry1Ac protoxin, while the SH-R strain presents approximately 2000-fold resistance to Btk formulation compared to

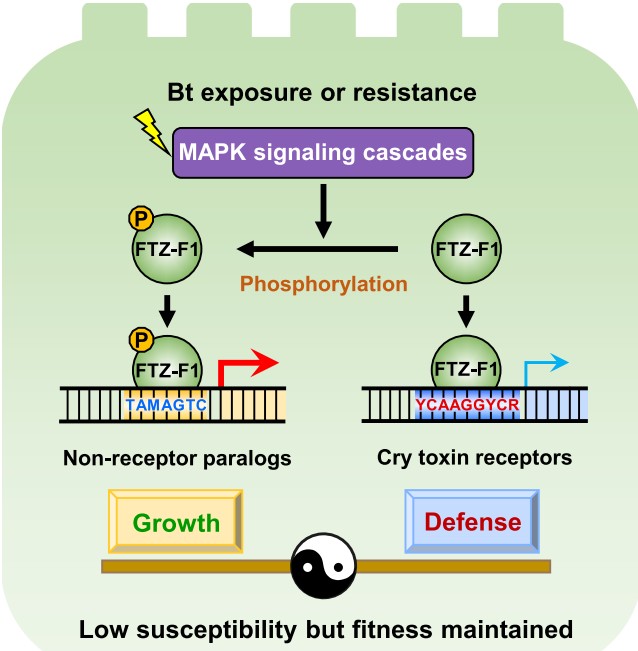

**Fig. 8 | Summary model of the MAPK-activated regulator FTZ-F1 maintaining a growth-defense balance with enhanced tolerance/resistance of *P. xylostella* to Bt Cry1Ac toxin.** Phosphorylated FTZ-F1 activates non-receptor genes via the motif "TAMAGTC", while non-phosphorylated FTZ-F1 induces receptor genes via the binding site "YCAAGGYCR". The activated MAPK cascade elevates the phosphorylation level of FTZ-F1, reducing the pool of non-phosphorylated TF, leading to upregulation of non-receptor gene expression and downregulation of receptor gene expression, which confers tolerance/resistance of *P. xylostella* to Cry1Ac toxin without growth penalty. Degenerate bases: M(A/C); Y(C/T); R(A/G).

the susceptible DBM1Ac-S strain. Larvae were fed with Jing Feng No. 1 cabbage (*Brassica oleracea* var. *capitata*) at 25 °C with 65% relative humidity (RH) and a 16:8 (light:dark) photoperiod, and adults were supplied with a 10% honey/water solution.

*Drosophila melanogaster* S2 cells were maintained in HyClone SFX-insect medium (HyClone) at 27 °C. *Spodoptera frugiperda* Sf9 cells were used to conduct the heterologous expression and subcellular localization experiments and were seeded on the coverslips with Sf-900 II SFM (Gibco) at 27 °C for 24 h.

### Toxin preparation and bioassay
The Cry1Ac protoxin used for toxicity bioassays was isolated from the Btk strain HD-73 and was purified and quantified[69]. A leaf-dip bioassay was performed to assess the toxic effect of Cry1Ac protoxin on *P. xylostella* larvae[66]. Ten third-instar larvae per group and three replicates were tested on each treatment of Cry1Ac toxin. Statistics on larval mortality was then conducted at 72 h post-treatment.

### RNA extraction, cDNA synthesis, and gDNA isolation
Different tissue samples of *P. xylostella* were dissected from fourth-instar larvae in ice-cold insect Ringer's solution (130 mM NaCl, 0.5 mM KCl, 0.1 mM CaCl₂). Total RNA was extracted using TRIzol reagent (Invitrogen). Subsequently, the first-strand cDNA for gene cloning was prepared using the PrimeScript II 1st strand cDNA Synthesis Kit (TaKaRa), and that for qPCR detection was synthesized using the PrimeScript RT kit (containing gDNA Eraser, Perfect Real Time) (TaKaRa) according to the manuals. Genomic DNA (gDNA) samples for promoter cloning were isolated from fourth-instar larvae using a TIANamp Genomic DNA Kit (TIANGEN) according to the manufacturer's instructions. All the prepared cDNA and gDNA samples were used immediately or stored at −20 °C until used.

### Cloning of the promoters and TFs
Specific PCR primers (Supplementary Table 4) were designed to amplify the promoters of multiple midgut genes viz. *APN1*, *APN3a*, *APN5*, *APN6*, *ABCB1*, *ABCC1*, *ABCC2*, *ABCC3*, and *ABCG1*, based on the 5′-untranslated region (5′-UTR) sequences of these genes in the *P. xylostella* genome dataset of LepBase (http://ensembl.lepbase.org/Plutella_xylostella_pacbiov1/) and the DBM-DB (http://116.62.11.144/DBM/). The coding sequences (CDSs) of *FTZ-F1*, *PREB*, and *RB1CC1* in *P. xylostella* were retrieved from the GenBank database (https://www.ncbi.nlm.nih.gov/) (*FTZ-F1*, XM_011551406 and XM_038122297; *PREB*, XM_038120942; *RB1CC1*, XM_038121053). These CDSs were further corrected using our previous *P. xylostella* midgut transcriptome database[70]. The gene-specific primers (Supplementary Table 4) were then designed by the Primer Premier 5.0 software (Premier Biosoft) to amplify their full-length CDSs. PCR amplification was performed using PrimeSTAR Max DNA Polymerase (TaKaRa) according to the manufacturer's protocol. The obtained amplicons were purified, subcloned into *pEASY*-T1 vectors (TransGen), and sequenced. The full-length cDNA sequences of both *αFTZ-F1* and *βFTZ-F1* genes from our *P. xylostella* strains have been deposited in the GenBank database (accession nos. MZ962431 and MZ962432). In addition, the coding sequences of FTZ-F1^T288A(-P) and FTZ-F1^T288D(P) were generated by gene synthesis (TsingKe) (Supplementary Table 5).

### Bioinformatic analysis
The amino acid sequences of TFs were deduced using the ExPASy translate tool (https://web.expasy.org/translate/). Analysis and alignment of DNA and protein sequences were performed using DNAMAN 9.0 (Lynnon BioSoft). Multiple sequence alignment was carried out using Clustal Omega (http://www.ebi.ac.uk/Tools/msa/clustalo/), and the results were further formatted using the GeneDoc 2.7 software (http://genedoc.software.informer.com/2.7/). Putative TFBSs in promoters were predicted using the JASPAR database (http://jaspar.genereg.net) and PROMO (http://alggen.lsi.upc.es/cgi-bin/promo_v3/promo/promoinit.cgi?dirDB=TF_8.3). The DNA binding motifs of FTZ-F1 were displayed using WebLogo 3 (http://weblogo.threeplusone.com/). The conserved domains of FTZ-F1 protein were analyzed by the Conserved Domain Database (CDD) at NCBI (https://www.ncbi.nlm.nih.gov/cdd/). The phylogenetic tree of FTZ-F1 proteins in various insects (Supplementary Table 6) was generated using MEGA 7.0 software with the neighbor-joining (NJ) method following the p-distance model, and 1000 bootstrap replicates. The scores of predicted phosphorylation sites were calculated by the disorder-enhanced phosphorylation predictor (DEPP) (http://www.pondr.com/cgi-bin/depp.cgi).

### Dual-luciferase assay
Promoter fragments were ligated into the firefly luciferase reporter vector pGL4.10 (Promega). In addition, promoters with mutant FBS and FBS^P were prepared by gene synthesis (TsingKe) (Supplementary Table 5). The CDSs of TFs were cloned into the pAc5.1/V5-His B (hereinafter called "pAc5.1") expression vector (Invitrogen) and, in doing so, were fused to a C-terminal V5-His tag. All primers for vector construction are given in Supplementary Table 4. The pGL4.73 vector (Promega) containing a *Renilla* luciferase gene was used as an internal control.

S2 cell line has been widely used in various non-model organisms to perform dual-luciferase reporter assays, thus, we conducted the dual-luciferase assays in S2 cells[62]. In brief, transfection of plasmids into S2 cells was performed using Lipofectamine 2000 transfection reagent (Thermo Fisher Scientific). The promoter constructs (600 ng) and the pGL4.73 plasmid (200 ng) were co-transfected into S2 cells to detect promoter activity, and the empty pGL4.10 vector was used as a control. TF expression plasmids (600 ng), promoter constructs (200 ng), and the pGL4.73 vector (100 ng) were co-transfected into S2 cells to validate the regulatory effects of TFs on the promoter, and the

empty pAc5.1 plasmid was used as a control. At 48 h post-transfection, luciferase activity was measured on a GloMax 96 Microplate Luminometer (Promega) by a Dual-Luciferase Reporter Assay System (Promega) according to the manufacturer's protocol. The relative luciferase activity (firefly luciferase activity/*Renilla* luciferase activity) of each construct was normalized to that of the control group. Each experiment was performed with three independent replicates.

## Y1H assay

The yeast one-hybrid (Y1H) assay was performed[62]. Briefly, bait plasmids were generated by inserting three tandem repeats of the wild-type FBS from the *ABCC2* promoter (5′-CTGTCCTGTAA−3′), its mutant FBS (FBS-M) (5′-CGCACACACGT−3′), wild-type FBS$^P$ from *ABCC1* promoter (5′-GTACAGTCA-3′), or its mutant FBS$^P$ (5′-GGCTCCGAAC-3′) into the pAbAi vector, the plasmids were then integrated into Y1HGold yeast. Subsequently, the minimum inhibitory concentrations of aureobasidin A (AbA) for normal growth of the bait strains were determined. Prey plasmids were generated by fusing the coding sequences of αFTZ-F1$^{T288A(-P)}$ and αFTZ-F1$^{T288D(P)}$ into the pGADT7 vector, which were then introduced into the bait strains and selected on SD/-Leu medium with AbA. The yeast co-transformed with the pGADT7-p53 and pAbAi-p53 plasmids was used as a positive control. The yeast co-transformed with the empty vector pGADT7, and the bait plasmid was used as a negative control.

## EMSA

The N-terminally His-tagged recombinant αFTZ-F1$^{T288A(-P)}$ and αFTZ-F1$^{T288D(P)}$ were expressed in *Escherichia coli* strain BL21 and purified using a His-tag Protein Purification Kit (Beyotime Biotechnology). Oligonucleotide probes for the wild-type FBS and FBS$^P$, mutant FBS, and FBS$^P$ that were labeled with biotin at the 5′-terminus were prepared by gene synthesis (TsingKe) (Supplementary Table 5). Before electrophoresis, the DNA fragment and purified proteins were incubated at 25 °C for 30 min. The DNA–protein complexes were then electro-transferred and detected with a LightShift Chemiluminescent EMSA Kit (Thermo Fisher Scientific) following the manufacturer's instructions. Blots were detected on the Tanon-5200 Chemiluminescent Imaging System (Tanon).

## Protein extraction and western blot

Midgut tissues were dissected from fourth-instar larvae in different strains. The tissues were homogenized in CelLytic M Cell Lysis Reagent (Sigma-Aldrich) supplemented with the EDTA-Free Complete Protease Inhibitor Cocktail (Roche) and the PhosSTOP Phosphatase Inhibitor Cocktail (Roche), and then were centrifuged to collect the supernatants. The nucleoproteins were extracted by the Nuclear and Cytoplasmic Protein Extraction Kit (Beyotime Biotechnology) according to the specification. Protein concentration was quantified with the Bradford assay (Biomed) according to the instructions. The obtained midgut protein samples were used immediately or stored at −80 °C until used.

For western blots, the prepared midgut proteins were mixed with protein loading buffer (CWBIO) and separated on 10% SDS-PAGE (CWBIO), then transferred to PVDF membranes (Merck Millipore). The membranes were further blocked with Bløk-PO buffer (Merck Millipore) and incubated with anti-FTZ-F1 polyclonal antibody (1:10000, produced by immunizing rabbits with Protein A/G-purified with the help of AtaGenix Laboratories, Wuhan, China) at 4 °C overnight, and then incubated with HRP-conjugated goat anti-rabbit IgG (1:5000, CWBIO) at 25 °C for 1 h. Blots were detected by the Tanon-5200 Chemiluminescent Imaging System (Tanon) using the SuperSignal West Pico Chemiluminescent reagent (Thermo Fisher Scientific). The images were analyzed using the ImageJ 1.51 software (https://imagej.nih.gov/ij/) by densitometry.

Phos-tag SDS-PAGE is an electrophoresis technique capable of simultaneous detection of phosphorylated and non-phosphorylated proteins by their band shift differences using a general antibody,

without the need to prepare an anti-phospho antibody. Western blot analysis with Phos-tag gels to separate phosphorylated from non-phosphorylated FTZ-F1 was performed following the handbook supplied by WaKo Co., Ltd. The Phos-tag SDS-PAGE gel was prepared by adding an additional 100 μm MnCl$_2$ and 50 μm Phos-tag (Wako) to the 10% SDS-PAGE. After loading the samples, the specific gel was run at 40 V overnight on ice and was washed twice by gently shaking in transfer buffer containing 1 mmol/L EDTA (CWBIO) for 10 min, and was then incubated in transfer buffer for another 20 min. The subsequent western blot analysis of the Phos-tag gel was performed as described above. Both the β-actin (1:2000, Abcam) and the Histone 3 (1:2000, ABclonal) were used as internal loading controls.

## MAPK inhibitor assays

To explore the effect of p38, JNK, and ERK on FTZ-F1, the resistant NIL-R larvae were treated with 30 μM of the specific inhibitors: SB203580 (Merk Millipore) for p38, SP600125 (Merk Millipore) for JNK and PD0325901 (TargetMol) for MEK1/2. The optimal inhibitors and their treatment concentrations and time had been optimized[27]. Inhibitor assays were conducted by a leaf-dip method similar to the toxicity bioassay. The inhibitors were dissolved in DMSO (Sigma-Aldrich) as stock solutions, which were then mixed with 0.05% (v/v) Triton X-100 solution. Leaf disks (10 cm in diameter) were soaked in the dissolved inhibitors or DMSO solution alone (as control). Thirty fourth-instar NIL-R larvae were fed on these leaf disks after air-drying. Midgut tissue was dissected at 6 h post-treatment to prepare RNA samples for qPCR analysis and protein samples for western blot.

## qPCR analysis

Gene expression levels were detected by real-time quantitative PCR (qPCR) analysis[67,71] using the specific primers listed in Supplementary Table 4. Briefly, the detection was conducted on the QuantStudio 3 Real-Time PCR System (Applied Biosystems) using FastFire qPCR PreMix (SYBR Green) (TIANGEN) according to the manufacturer's instructions. Relative expression levels were calculated using the $2^{-\Delta\Delta Ct}$ method and normalized to the level of the internal control ribosomal protein *L32* (*RPL32*) gene (GenBank accession no. AB180441).

## RNA interference

RNAi-induced silencing of *MAP4K4* and *FTZ-F1* were performed to explore the in vivo regulatory relationships among the *MAP4K4*, *FTZ-F1*, and multiple midgut genes in *P. xylostella*[67]. The specific dsRNA was synthesized using the T7 Ribomax Express RNAi System (Promega). The gene-specific primers (Supplementary Table 4) for dsRNA synthesis were designed for the gene-specific region to avoid potential off-target effects, and no specific hit to other homologous genes was detected by BLASTN searches of GenBank and the *P. xylostella* genome database, further validating the specificity of the selected dsRNA fragments. Then, microinjection of a sub-lethal dose of dsRNA (100 ng for dsFTZ-F1, 300 ng for dsMAP4K4) was carried out in newly molted third-instar *P. xylostella* larvae using the Nanoliter 2000 microinjection system (World Precision Instruments). Silencing effects were tested at 48 h post-injection by qPCR and a subsequent 72 h leaf-dip bioassay.

## Subcellular localization

The recombinant EGFP-FTZ-F1 fusion (with EGFP fused to the N-terminus of FTZ-F1) coding plasmid was transfected into Sf9 cells using FuGENE HD (Promega) at a ratio of 1:3 (plasmids to FuGENE). The transfected cells were fixed with 4% paraformaldehyde (w/v, PFA) for 15 min at 48 h post-transfection, and then permeabilized with 0.5% Triton X-100 for 20 min. After three washes with PBS, the nuclei were stained with 5 μM DAPI (Abcam) for 15 min at room temperature. Non-transfected cells were used as a negative control and cells transfected with the Pie-EGFP-N1 vector were used as a positive control. Images

were visualized with a Leica laser scanning confocal microscope (Leica, TCS SP8, Wetzlar, Germany).

## Immunoprecipitation and LC-MS/MS assay

Sf9 cells transfected with EGFP-FTZ-F1 fusion protein were harvested and lysed in lysis buffer (20 mM Tris-HCl [pH 7.5], 100 mM KCl, 2 mM MgCl$_2$, 0.3% IGEPAL CA-630, 1 mM protease inhibitor cocktail and 1 mM phosphatase inhibitor cocktail (Roche)) on ice. Lysed total protein samples were preincubated with protein A/G beads on a rotating wheel at 4 °C for 1 h. The beads were removed, and the protein was mixed with 5 µg anti-GFP (Abcam) or 4 µg anti-IgG (Sigma-Aldrich) overnight at 4 °C, and subsequently incubated with protein A/G beads again for 3 h at 4 °C. Beads were pelleted on a magnetic stand and the supernatant was discarded. The beads were then washed five times with lysis buffer. Elution was performed by adding SDS loading buffer followed by incubation at 95 °C for 10 min. Immunoprecipitation was applied for western blot and LC-MS/MS assays.

The eluent was digested with trypsin enzyme (Promega) following the filter-aided sample preparation (FASP) protein digestion protocol. LC-MS/MS experiments were carried out with an Orbitrap Fusion Lumos Tribrid mass spectrometer (Thermo Fisher Scientific) coupled to an EASY-nLC 1200 system (Thermo Fisher Scientific). The analytical columns (75 µm × 25 cm, 5 µm, 100 Å, C18) were equilibrated in buffer A (0.1% formic acid). The digested peptides were automatically injected onto the EASY trap column (100 µm × 2 cm, 5 µm, 100 Å, C18) (Thermo Fisher Scientific) and then separated using the following gradient of buffer B (0.1% formic acid acetonitrile) at a 200 nl/min flow rate: 0–40 min, 5–28% buffer B, 40–42 min, 28–90%, 42–60 min, hold at 90%. The hydrolysates were desalted and separated by capillary high-performance liquid chromatography and analyzed by an Orbitrap Fusion Lumos Tribrid mass spectrometry. The scan analysis lasted 60 min and the master scans were acquired at a resolution of 120,000 at $m/z$ 200, the scan range of 375–1800 $m/z$, top speed, AGC target of 4e$^5$, maximum IT of 50 ms, number of scan ranges of 1, dynamic exclusion of 40.0 s. The data-dependent mode was cycle time and the time between the master scan was 3 s. MS2 scan was performed by HCD fragmentation with a resolution of 50,000 at $m/z$ 200, maximum IT of 105 ms, AGC target of 1e$^5$, microscans of 1. All results were analyzed by applying Proteome Discoverer 2.4 software (Thermo Fisher Scientific). The search parameters were as follows: trypsin digestion with two missed cleavages was permitted, charge states 1$^+$, 2$^+$, 3$^+$ for precursor ion. The mass errors of precursor ion and fragment ions were 10 ppm and 0.05 Da.

## CRISPR/Cas9 experiment

CRISPR/Cas9-mediated single knockout of *ABCC1*, a double mutant of *APN5* and *APN6*, as well as the triple mutations of *ABCC1*, *APN5*, and *APN6* were performed to elaborate the in vivo important roles of non-receptor genes and their interactions, as reported elsewhere[37,72]. Briefly, three optimal sgRNAs targeting *ABCC1*, *APN5*, and *APN6* genomic sequences were designed (Supplementary Table 4) and the potential off-target effects of all the sgRNAs were eliminated. For double-gene knockout, about 1 nl mixture of two sgRNAs and Cas9 protein (the final concentration of each sgRNA and Cas9 protein was 100 ng/µl) were simultaneously microinjected into individual eggs from the resistant NIL-R strain. According to the adjacent structure of the *APN5* and *APN6* genes in the DBM genome, four gene-specific primer pairs were designed to determine the mutagenesis of the double-gene regions according to PCR banding pattern of the resultant amplicons and DNA direct sequencing (Supplementary Table 4). For single or triple mutations, a mixture of Cas9 protein (200 ng/µl) and sgRNA (150 ng/µl) was injected into individual eggs from the resistant NIL-R or double-mutant strain (N6-5KO), respectively. In addition, a nondestructive genotyping method was applied to all mutation types, i.e., the gDNA samples were extracted from exuviates of individual fourth-instar *P. xylostella* larvae as templates to amplify the DNA

fragment surrounding the sgRNA target site for DNA sequencing. Subsequently, the stable homozygous mutant strains were constructed by mutation screening and germline transformation strategy (Supplementary Tables 1, 2).

## Heterologous expression

Heterologous expression of *ABCC1-3* genes was performed[14,26]. The full-length cDNA sequences of *ABCC1-3* genes were cloned from the susceptible DBM1Ac-S *P. xylostella* larvae and inserted into the pie2-EGFP-N1 expression vector to generate three recombinant plasmids (pie2-EGFP-ABCC1/ABCC2/ABCC3) (Supplementary Table 4). All the recombinant plasmids containing the pie2 promoter and the EGFP-ABCC fusion proteins with the plasmid which only expressed EGFP protein as a control. Subsequently, the recombinant ABCC1-3 proteins were transiently expressed in vitro in Sf9 cells. The specific interaction between all the recombinant proteins and Cry1Ac toxin was determined by immunolocalization in Sf9 cells after transfection. The transfected Sf9 cells were incubated with trypsin-activated Cry1Ac toxin (100 mg/L) and fixed in 4% paraformaldehyde. After blocking, the cells were respectively incubated with primary rabbit polyclonal anti-Cry1Ac antibody (1:100, produced in our lab) and goat anti-rabbit secondary antibody conjugated with Alexa Fluor 555 (1:500, Abcam). The treated cells were observed under the LSM 700 confocal microscope (Carl Zeiss) equipped with the ZEN 2012 software (Carl Zeiss). A CCK-8 (WST-8 in the Cell Counting Kit-8, Dojindo) assay was performed to detect cytotoxicity. The absorbance was measured at 450 nm after 24 h incubation with Cry1Ac toxin, and the proportion of viable cells (single or combined transfection) was measured relative to untreated Sf9 cells, which was set as 100%.

## Fitness cost analysis

A series of physiological parameters in *P. xylostella* mutant strains were compared to analyze the fitness cost induced by the CRISPR knockouts, the susceptible DBM1Ac-S and the parental NIL-R resistant strains were used as controls. The biological parameters measured were pupal morphology, pupation percentage, pupal weight, pupation duration, and hatching rate. Ten second-instar larvae from each strain were kept on fresh cabbage leaves without exposure to any Bt toxin, and each test was replicated five times.

## Statistical analyses and data visualization

For dual-luciferase assays, qPCR, western blot, bioassay data, and fitness costs, significant differences between different groups were evaluated by one-way ANOVAs with Tukey's test using IBM SPSS Statistics 23.0 (https://www-01.ibm.com/support/docview.wss?uid=swg24038592). Graphs were constructed by Microsoft Office 2010 (https://www.microsoft.com/en-us/microsoft-365/previous-versions/office-2010), SigmaPlot 12.5 (https://systatsoftware.com/products/sigmaplot/), or GraphPad Prism 8.3 (https://www.graphpad.com/scientific-software/prism/).

## Reporting summary

Further information on research design is available in the Nature Research Reporting Summary linked to this article.

# Data availability

The full-length cDNA sequences of all the cloned genes in this study have been deposited in the GenBank database under accession numbers MZ962431 and MZ962432. The gene or genome databases including DBM-DB (http://116.62.11.144/DBM/), LepBase (http://ensembl.lepbase.org/Plutella_xylostella_pacbiov1/), and GenBank (https://www.ncbi.nlm.nih.gov/) were used to obtain sequences of target genes and their promoters. The authors declare that the data supporting the findings of this study are available within the paper and its Supplementary Information. The source data underlying Figs. 1, 2,

3a, c, d, g, h, 4a, b, 5, 6, 7c–f and Supplementary Figs. 2, 5, 6, 9b, 10b are provided as a Source Data file. Source data are provided with this paper.

## Code availability

No custom code or algorithms were used in this study.

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

## Acknowledgements

This work was supported by the Laboratory of Lingnan Modern Agriculture Project (NT2021003), the National Natural Science Foundation of China (32022074; 32172458; 32221004), the China Postdoctoral Science Foundation (2022TQ0369), the Beijing Key Laboratory for Pest Control and Sustainable Cultivation of Vegetables and the Science and Technology Innovation Program of the Chinese Academy of Agricultural Sciences (CAAS-ASTIP-IVFCAAS).

## Author contributions

Z.G., L.G., J.Q., N.C., and Y.Z. designed the research. Z.G., L.G., J.Q., F.Y., D.S., Q.W., and S.W. performed the experiments. Z.G., L.G., and J.Q. analyzed the data. Z.G., L.G., J.Q., N.C., X.Z., A.B., M.S., and Y.Z. wrote and revised the manuscript.

## Competing interests

The authors declare no competing interests.
