## [Peer Review File · Nature Communications]

Reviewers' Comments:

Reviewer #1:

Remarks to the Author:

This paper reports the finding that a single transcription factor, Ftz-F1, provides resistance to Bt toxin by coordinately up- and down-regulating expression of distinct sets of receptors. Studies were done to compare sensitive and resistant lines of *Plutella xylostella*, diamondback moth. The authors previously showed that candidate Bt toxin receptors are downregulated in resistant strains while other family members that do not function as Bt receptors are upregulated. This cross regulation provides for compensation for other, detrimental impacts of receptor downregulation, and thus this represents a potential mechanism for evolved resistance without loss of fitness. In the current study, the authors show that Ftz-F1 coordinately regulates this gene set and that it differentially up- or down-regulates specific genes depending on its phosphorylation state, with different phospho-forms having different DNA binding specificity.

This is an interesting and important paper. It is clear and well-written. The range of experiments is extensive. The data are convincing – I see no experimental flaws – and the conclusions are well supported by the results. This makes for a very tight story to explain rapid evolution of pesticide resistance for an important pest species that may well have parallels for other pest species. In addition, it addresses mechanisms of action of a TF that is highly conserved, including in mammals, suggesting the importance of phosphorylation in target site recognition. Thus, this work will also be of interest to those studying transcription regulation in a range of organisms.

I suggest the following changes, none of which require additional experiments:

1. The changes in expression shown in Fig. 1 and 3 are in the range of ~ 2 fold. This is not a huge change. I ask that the authors comment on this relative to other publications in FTZ-F1 and/or other TFs using the same assay.
2. For the FBS, the numbering should be relative to transcription start site since the translation start site is not relevant to this study (e.g., a translation start site could reside in any exon, at any distance from the promoter, since the 5'UTR length can vary, all of which is not relevant to TF binding). There is a large intron downstream of *alphaftz-f1* exon 1 – are these sites in the intron or upstream of exon 1? The text cannot refer to a promoter, unless TSS was mapped
3. The authors show that expression is highest in adult females – is that the life stage impacted by Bt? It would be interesting to hear the authors' thoughts about how selection on *ftz-f1* led to resistant populations.
4. The authors use multiple techniques that demonstrate quite striking differences in gene regulation and DNA binding between these two forms of the protein (phospho-mimic vs. non-phospho mimic). The T288 residue appears to be in the hinge regions between the DBD and LBD. How do the authors explain the large change in DNA binding based on phosphorylation of this residue? The authors do cite one other example of this type of effect but it is unusual enough that the finding warrants more discussion with respect to this particular protein as well as findings from others.
5. Fig. 7 shows that the p38 treatment led to decreased phosphoFtz-F1. This change is suggestive but less convincing than other figures in the paper and is tangential to the main story line. I would move it to a supplementary figure or remove it altogether.
6. Several experiments use tagged versions of FTZ-F1: His-tag for DNA binding assays; EGFP tags for other assays. I could not tell if these were N- or C-terminal tags but C-terminal tags could impact the AF-2 domain (although the AF-2 may not be at the very C-terminus of the protein, which it is in *Drosophila*) and thereby could impact transcription activation. Please comment on potential impacts of the TAGS and how that was addressed.
7. The EMSA would be easier to understand if shown as a single gel, which I think it was with the last 2 lanes looking the same for each part of the figure.
8. Replace "normal" with "wild type" when referring to growth as well as "wild type" or "consensus" when referring to sequences

Reviewer #2:

Remarks to the Author:

It is very important for the adaptive evolution of organisms to reduce the fitness cost as much as possible while effectively resisting pathogen infection or toxic substance exposure. Guo et al

revealed that non-phosphorylated and phosphorylated forms of a single MAPK-modulated transcription factor FTZ-F1 can respectively orchestrate down-regulation of Bt toxin receptors and up-regulation of non-receptor paralogs via two distinct binding sites thereby presenting Bt resistance without growth penalty. This is the first report on the molecular mechanisms of growth-defense trade-offs in insect pests, which would be of great interest to a broad of readers. The whole experiments were designed logically, the conclusion is strongly supported by the data, and the manuscript is well written. It is a perfect job and is acceptable as it is.

Line 290: delete difference.

Reviewer #3:

Remarks to the Author:

In this paper, the disappearance of fitness cost of Bt resistance in *Plutella xylostella* was studied. Through a series of work, it seems that phosphorylation of a transcription factor FTZ-F1 determines Bt resistance and compensation of fitness costs. This mechanism is that, the insect can regulate phosphorylation of transcription factors FTZ-F1 through MAPK pathway. And the up-regulation of phosphorylation levels of FTZ-F1 just decrease the expression of Bt receptor genes and induce expression of non-receptor homologous genes, which compensates the fitness caused by the down-regulated expression of receptor genes. It sounds like an interesting and very accidental story.

It is a big work. resistance (including, Bt toxins, insecticides, and any other environmental factors), in particular high level resistance through regulatory pathways often comes with a high fitness cost that requires a range of physiological regulation by the organism to compensate for. It's hard to imagine that a change in just one gene, whether it's phosphorylation or something else, could bring about such a perfect fit. It is hard to imagine that the cost of fitness could be completely eliminated, even if it could be partially remedied. This should be deeply discussed in the discussion.

One of the key methods in this paper is cell experiment. The differences between *Drosophila* cells and diamondback moth cells are too great to be convincing. The reliability of simultaneous expression of transcription factors and promoters in vitro is questionable. The authors should use other *Lepidoptera* cells to replace the *Drosophila* cells, and add some vivo experiments.

Other comments :

1. Ftz-f1 is just an acronym. I didn't find the full name.
2. Figure 2 should be moved to the supplementary as it reflects too little information to be placed on the main image.

Reviewer #4:

Remarks to the Author:

The manuscript has two different aims (i): characterized a transcription factor (TF) that they hypothesized can be involved in the regulation of the expression of different genes (Bt toxin receptor-encoding genes) and (ii) to determine if the reduction of the TF affects the susceptibility of Cry1Ac (associates with Cry1Ac resistance).

The results showed that the TF (FTZ-F1) has two forms, non-phosphorylated and phosphorylated that can regulate the expression of Bt toxin receptors and non-receptor paralogs. Moreover, the authors identify the functional binding sites on the promoters of the genes studied, observing a non-canonical binding site for the non-receptor genes. Also, they demonstrate that both FTZ-F1 forms preferentially bind to different binding domains supporting the hypothesis that this ability would help to regulate the expression levels of different genes.

On the other hand, they successfully showed that MAPK cascade is playing a role in the regulation of the phosphorylation of FTZ-F1. And interestingly, they showed that non-receptor paralogs are important for maintaining fitness.

The authors performed a massive work and they obtained very nice and cute results. In general,

the article is very interesting and important, however, there are issues that should be addressed by the authors. My revised suggestions and comments are listed as follows:

- Results section

- I will suggest moving lines from 122 to 126 to the next section. The authors make a brief introduction about the need to find the binding sites of the TF at the end of this section, and they give in advance part of the results that are going to be shown in depth in the following section.

- In figure 1, move panel b to supplemental material or add to figure 2. Also, in the figure legend is written "Potential FTZ-F1 binding sites (FBS) predicted in resistance-related genes are depicted by yellow ellipses" I suggest modifying the term "resistance-related genes" as not all the genes have been related to resistance. In the manuscript, the authors use other terms such as receptor-encoding genes and non-receptor paralogs that I think it is more appropriate.

- In the section "Phosphorylated and non-phosphorylated FTZ-F1 preferentially bind to different DNA motifs" Only two mutated FTZ-F1 forms altered the expression of the genes studied. So, I have two comments, (i) Do you know why the other mutants do not modify the expression of the genes? Any suggestion or explanation, (ii) The results (Fig 4d) indicate that in the first experiment the S2 cells produced both the phosphorylated and the non-phosphorylated FTZ-F1. For that reason, all the genes (receptor and non-receptors) had affected their promotor activity, isn't it? Did the authors check the presence of both forms in the first experiment? To avoid misunderstanding this observation needs to be explained in the manuscript.

- Lines 210-213, Why do the authors check if the TFs are located in the nucleus at this stage and not before? It was quite surprising to find this result (subcellular localization) in this section and not at the beginning of the manuscript. Furthermore, why this confirmation was not done using the S2 cells? If I am right, the luciferase assays were done in these cells.

- Section: "The level of phosphorylated FTZ-F1 associates with Cry1Ac resistance",

o Is the antibody used able to recognize one or the two isoforms of the FTZ-F1? I think it is interesting to know this information.

o Why the authors did not include the b-actin as a control in fig 6b? The authors claim that "The level of phosphorylated FTZ-F1 was found to be significantly higher in all four resistant strains compared to the susceptible DBM1Ac-S strain" Any statistics analysis has been done to state that? Or maybe it is only a qualitative observation. Again, Why the b-actin was not used as a control in this experiment?

o I totally agree that the silencing of FTZ-F1 in larvae of the resistant strain NIL-R was accompanied by a reduction in both phosphorylated and non-phosphorylated FTZ-F1. But I do not agree with your last conclusion: "These data support the conclusion that the differential phosphorylation of FTZ-F1 in vivo is associated with Cry1Ac resistance in *P. xylostella*". From my point of view, the results presented in this section suggest that lower levels of FTZ-F1 were accompanied by a decrease in susceptibility Cry1Ac but you cannot conclude anything about the role of the differential phosphorylation.

- Section: "The MAPK cascade regulates the phosphorylation level of FTZ-F1"

o Did you test the susceptibility of the larvae silenced of MAP4K? it would be nice to know the effect of silencing the map4k4 gene, to know if your predictions are correct or not.

o Fig 7 pane c and f, Are these values significant?

- Section: Non-receptor genes are conducive to fitness in resistant *P. xylostella*

o I will suggest moving lines 292 to 296 to the discussion section.

o Lines 296-299: "Ectopic expression and cytotoxicity experiments presented here show that Sf9 cells expressing ABCC2 or ABCC3 (but not ABCC1) can bind and be susceptible to Cry1Ac toxin (Supplementary Fig. 8a, b). These results confirm the distinction between functional receptors and their non-receptor paralogs". From my point of view, this information is not needed here. I would say that is distracting from the important information.

o I think that results from this section (lanes 300 to 309) should be more emphasized as they are really relevant for this study. As I commented before, in this section there is too much info that can be moved or deleted because it is distracting from the real topic.

Response to Reviewers' comments

Pilar Morera Margarit
Associate Editor
Nature Communications

August 8, 2022

Dear Editor,

We would like to thank you, and the four anonymous expert reviewers, for the time spent reviewing our manuscript (Manuscript Number: NCOMMS-22-19835) entitled “A single transcription factor facilitates an insect host combating *Bacillus thuringiensis* infection while maintaining fitness” submitted to *Nature Communications*. The constructive comments from the four reviewers have been instrumental in improving the manuscript. Moreover, we have also revised the manuscript according to the *Nature Communications* formatting instructions. Changes in the revised manuscript have been highlighted in yellow. We hope you will find the new version of the manuscript suitable for publication.

The following is a point-by-point response to the reviewer's comments:

Responses to Reviewers' Questions

Reviewer #1 (Remarks to the Author):

This paper reports the finding that a single transcription factor, Ftz-F1, provides resistance to Bt toxin by coordinately up- and down-regulating expression of distinct sets of receptors. Studies were done to compare sensitive and resistant lines of Plutella xylostella, diamondback moth. The authors previously showed that candidate Bt toxin receptors are downregulated in resistant strains while other family members that do not function as Bt receptors are upregulated. This cross regulation provides for compensation for other, detrimental impacts of receptor downregulation, and thus this represents a potential mechanism for evolved resistance without loss of fitness. In the current study, the authors show that Ftz-F1 coordinately regulates this gene set and that it differentially up- or down-regulates specific genes depending on its phosphorylation state, with different phosphor-forms having different DNA binding specificity.

This is an interesting and important paper. It is clear and well-written. The range of experiments is extensive. The data are convincing – I see no experimental flaws – and the conclusions are well supported by the results. This makes for a very tight story to explain rapid evolution of pesticide resistance for an important pest species that may well have parallels for other pest species. In addition, it addresses mechanisms of action of a TF that is highly conserved, including in mammals, suggesting the

importance of phosphorylation in target site recognition. Thus, this work will also be of interest to those studying transcription regulation in a range of organisms.

We appreciate the reviewer's positive comments on our work.

I suggest the following changes, none of which require additional experiments:

1. The changes in expression shown in Fig. 1 and 3 are in the range of ~ 2 fold. This is not a huge change. I ask that the authors comment on this relative to other publications in FTZ-F1 and/or other TFs using the same assay.

We agree that the fold changes are modest - similar changes have been reported with this TF and we have added text to the manuscript (Lines 114-116) citing another example. We would also like to point out that the fold differences observed for the midgut genes are in line with the data that we got previously when components of the MAPK signaling pathway were knocked down using RNAi (Guo et al., PLoS Pathog., 2021, 17: e1009917).

2. For the FBS, the numbering should be relative to transcription start site since the translation start site is not relevant to this study (e.g., a translation start site could reside in any exon, at any distance from the promoter, since the 5'UTR length can vary, all of which is not relevant to TF binding). There is a large intron downstream of alphaftz-fl exon 1 – are these sites in the intron or upstream of exon 1? The text cannot refer to a promoter, unless TSS was mapped.

The use of the translational start site provides a single, fixed, reference point to enable the mapping of the various sequence elements to the DNA sequence, and so we would prefer to stick with this system. While the DNA sequence encoding the translational start site can be predicted with some certainty from the literature, the transcriptional start site(s) would have to be mapped anew. The luciferase assays in Fig. 1 strongly indicate that the 5'-UTR regions that we used contain the promoter region. The large intron downstream of exon 1 is downstream of the referenced translational start site, and so all of the identified FBSs are upstream of the first exon.

3. The authors show that expression is highest in adult females – is that the life stage impacted by Bt? It would be interesting to hear the authors' thoughts about how selection on ftz-fl led to resistant populations.

It is actually the larval stage that is affected by Bt, and there is no obvious sex bias. FTZ-F1 is most likely to be involved in controlling a number of different physiological processes in the insect, and perhaps one or more of these are highly regulated in female adults. We presume that there is some form of scaffold system that allows the MAPK pathway and FTZ-F1 to specifically target the midgut genes.

4. *The authors use multiple techniques that demonstrate quite striking differences in gene regulation and DNA binding between these two forms of the protein (phospho-mimic vs. non-phospho mimic). The T288 residue appears to be in the hinge regions between the DBD and LBD. How do the authors explain the large change in DNA binding based on phosphorylation of this residue? The authors do cite one other example of this type of effect but it is unusual enough that the finding warrants more discussion with respect to this particular protein as well as findings from others.*

Without detailed structural information, we can't propose a mechanism by which phosphorylation at T288 (which as pointed out is not part of the DNA binding motif) affects binding. However, we have provided in the text another example of a protein in which phosphorylation at a distal site influences DNA binding (Lines 408-411).

5. *Fig. 7 shows that the p38 treatment led to decreased phosphoFtz-F1. This change is suggestive but less convincing than other figures in the paper and is tangential to the main story line. I would move it to a supplementary figure or remove it altogether.*

Based on another reviewer's comments, we have performed some more experiments with the MAPK inhibitors. The combined data (now Fig. 6) strengthen this particular story line and so it is retained as a main figure. We provide this information in the revised manuscript (Lines 273-274).

6. *Several experiments use tagged versions of FTZ-F1: His-tag for DNA binding assays; EGFP tags for other assays. I could not tell if these were N- or C-terminal tags but C-terminal tags could impact the AF-2 domain (although the AF-2 may not be at the very C-terminus of the protein, which it is in Drosophila) and thereby could impact transcription activation. Please comment on potential impacts of the TAGS and how that was addressed.*

The proteins used in the EMSA experiments were N-terminally His-tagged, we have added this information to the Methods (Line 563). For the co-localization experiments (now Fig. 4), an N-terminal EGFP fusion was used, this information has also been added to Methods (Lines 644-645). The form of FTZ-F1 used in the co-transfection luciferase assay contained C-terminal V5+His tags (clarified in Methods, Line 530). Both of these are relatively small though and we believe that the impact is likely to be minimal. Activity associated with the C-terminally tagged protein (Fig. 1) was consistent with data obtained with the N-terminal tagged TF (Fig. 4).

7. *The EMSA would be easier to understand if shown as a single gel, which I think it was with the last 2 lanes looking the same for each part of the figure.*

The data shown in the new Fig. 4a and 4b are from two completely independent

experiments, we believe that it is better to present as two separate gels – one for FBS and the other for FBS^P.

8. Replace “normal” with “wild type” when referring to growth as well as “wild type” or “consensus” when referring to sequences

We have altered the manuscript based on this suggestion and hope that it is now clearer for the reader.

Reviewer #2 (Remarks to the Author):

It is very important for the adaptive evolution of organisms to reduce the fitness cost as much as possible while effectively resisting pathogen infection or toxic substance exposure. Guo et al revealed that non-phosphorylated and phosphorylated forms of a single MAPK-modulated transcription factor FTZ-F1 can respectively orchestrate down-regulation of Bt toxin receptors and up-regulation of non-receptor paralogs via two distinct binding sites thereby presenting Bt resistance without growth penalty. This is the first report on the molecular mechanisms of growth-defense trade-offs in insect pests, which would be of great interesting to a broad of readers.

The whole experiments were designed logically, the conclusion is strongly supported by the data, and the manuscript is well written. It is a perfect job and is acceptable as it is.

We are very grateful for the reviewer’s positive comments.

Lines 290: delete difference.

Corrected.

Reviewer #3 (Remarks to the Author):

*In this paper, the disappearance of fitness cost of Bt resistance in *Plutella xylostella* was studied. Through a series of work, it seems that phosphorylation of a transcription factor FTZ-F1 determines Bt resistance and compensation of fitness costs. This mechanism is that, the insect can regulate phosphorylation of transcription factors FTZ-F1 through MAPK pathway. And the up-regulation of phosphorylation levels of FTZ-F1 just decrease the expression of Bt receptor genes and induce expression of non-receptor homologous genes, which compensates the fitness caused by the down-regulated expression of receptor genes. It sounds like an interesting and very accidental story.*

It is a big work. resistance (including, Bt toxins, insecticides, and any other environmental factors), in particular high level resistance through regulatory

pathways often comes with a high fitness cost that requires a range of physiological regulation by the organism to compensate for. It's hard to imagine that a change in just one gene, whether it's phosphorylation or something else, could bring about such a perfect fit. It is hard to imagine that the cost of fitness could be completely eliminated, even if it could be partially remedied. This should be deeply discussed in the discussion.

The reviewer is not alone in being amazed that such a simple, but elegant, system exists that can bring about resistance with minimal fitness costs. We have long considered the possibility that we are looking at a normal physiological process that has serendipitously become a pathogen response/resistance one. We have been unable to come up with any sensible explanation and so have hypothesized that it is not an “accidental story” but instead the result of a very long history of interaction between this host and its pathogen. We have expanded our final conclusion a bit in the Discussion section (Lines 438-447).

One of the key methods in this paper is cell experiment. The differences between drosophila cells and diamondback moth cells are too great to be convincing. The reliability of simultaneous expression of transcription factors and promoters in vitro is questionable. The authors should use other Lepidoptera cells to replace the drosophila cells, and add some in vivo experiments.

S2 cell lines have widely been used to perform luciferase reporter assays for genes from various non-model organisms. For instance, they have been used with locusts for detecting the effect of miRNA on target genes (He et al., Proc. Natl. Acad. Sci. U. S. A., 2016, 113: 584–589). Even in penaeid shrimp, dual luciferase reporter assay in S2 cells have been used and recognized (Wang et al., Dev. Comp. Immunol., 2011, 35, 105–114). We agree that there will be differences between *Drosophila* cells and *Plutella* cells, however, using non-homologous cell lines can reduce potential endogenous interference, for example, the presence of non-recombinant FTZ-F1 capable of acting on the cloned promoter. Other cellular experiments were conducted in lepidopteran Sf9 cells, since here endogenous factors other than basic transcription/translation processes may be important. *In vivo* work has been performed to support the results from the cell assays (Fig. 5a-h, Fig. 6a-f, Fig. 7b-f).

Other comments:

1, *Ftz-f1 is just an acronym. I didn't find the full name.*

We apologize for this oversight. The full name of FTZ-F1 (Fushi tarazu factor 1) has been given in the revised manuscript (Lines 31, 83 and 95).

2, *Figure 2 should be moved to the supplementary as it reflects too little information*

to be placed on the main image.

We still believe that the data presented in Fig. 2 is part of the main story of identifying two distinct binding sites. Meanwhile, we have also realized that there was some duplication between Figs. 1 and 2. As a result, we have simplified and combined both figures (now Fig. 1). The original Fig. 1b has been moved to the supplemental material as the new Supplementary Fig. 3.

Reviewer #4 (Remarks to the Author):

The manuscript has two different aims (i): characterized a transcription factor (TF) that they hypothesized can be involved in the regulation of the expression of different genes (Bt toxin receptor-encoding genes) and (ii) to determine if the reduction of the TF affects the susceptibility of Cry1Ac (associates with Cry1Ac resistance).

The results showed that the TF (FTZ-F1) has two forms, non-phosphorylated and phosphorylated that can regulate the expression of Bt toxin receptors and non-receptor paralogs. Moreover, the authors identify the functional binding sites on the promoters of the genes studied, observing a non-canonical binding site for the non-receptor genes. Also, they demonstrate that both FTZ-F1 forms preferentially bind to different binding domains supporting the hypothesis that this ability would help to regulate the expression levels of different genes.

On the other hand, they successfully showed that MAPK cascade is playing a role in the regulation of the phosphorylation of FTZ-F1. And interestingly, they showed that non-receptor paralogs are important for maintaining fitness.

The authors performed a massive work and they obtained very nice and cute results. In general, the article is very interesting and important, however, there are issues that should be addressed by the authors.

We appreciate the reviewer's precise summary and positive comments.

My revised suggestions and comments are listed as follows:

- Results section

- I will suggest moving lines from 122 to 126 to the next section. The authors make a brief introduction about the need to find the binding sites of the TF at the end of this section, and they give in advance part of the results that are going to be shown in depth in the following section.

We totally agree to the reviewer's suggestion and have moved the relevant text to the beginning of the next section (Lines 128-132).

- In figure 1, move panel b to supplemental material or add to figure 2. Also, in the figure legend is written "Potential FTZ-F1 binding sites (FBS) predicted in

resistance-related genes are depicted by yellow ellipses” I suggest modifying the term “resistance-related genes” as not all the genes have been related to resistance. In the manuscript, the authors use other terms such as receptor-encoding genes and non-receptor paralogs that I think it is more appropriate.

We agree that there was duplication between Figs. 1 and 2, we therefore have simplified and combined both figures (now Fig. 1). The original Fig. 1b has been moved to the supplemental material as the new Supplementary Fig. 3 and the terminology in the legend has also been changed.

- In the section “Phosphorylated and non-phosphorylated FTZ-F1 preferentially bind to different DNA motifs” Only two mutated FTZ-F1 forms altered the expression of the genes studied. So, I have two comments, (i) Do you know why the other mutants do not modify the expression of the genes? Any suggestion or explanation, (ii) The results (Fig. 4d) indicate that in the first experiment the S2 cells produced both the phosphorylated and the non-phosphorylated FTZ-F1. For that reason, all the genes (receptor and non-receptors) had affected their promoter activity, isn't it? Did the authors check the presence of both forms in the first experiment? To avoid misunderstanding this observation needs to be explained in the manuscript.

Comment (i):

Although four sites on FTZ-F1 were found to be phosphorylated *in vivo*, we cannot infer that all four post-translational modifications are important. The other three could simply represent phosphorylation events that have no biological significance, or they are involved in a function of FTZ-F1 but unrelated to the system that we are studying.

Comment (ii):

To confirm our assumption, nucleoproteins from S2 cells (untransfected or transfected with recombinant FTZ-F1) were isolated and the phosphorylation status of the TF checked. As shown in the new Supplementary Fig. 6 and discussed in Lines 198-206 both forms of FTZ-F1 were indeed found.

- Lines 210-213, Why do the authors check if the TFs are located in the nucleus at this stage and not before? It was quite surprising to find this result (subcellular localization) in this section and not at the beginning of the manuscript. Furthermore, why this confirmation was not done using the S2 cells? If I am right, the luciferase assays were done in these cells.

We apologize for not explaining the reasoning for this experiment better. We were aware of the fact that altering the phosphorylation of a TF can alter its activity by removing it from the nucleus. Since our data had hypothesized that both phosphorylated and non-phosphorylated forms were active, we wanted to ensure that

they were both present in the nucleus. We have amended that section of the manuscript (Lines 221-224). We have explained the various use of S2 and Sf9 cells in our response to reviewer 3 as mentioned above.

- Section: *“The level of phosphorylated FTZ-F1 associates with CryI_{Ac} resistance”*.
o Is the antibody used able to recognize one or the two isoforms of the FTZ-F1? I think it is interesting to know this information.

We adopted an unconventional method, called Phos-tag SDS-PAGE, to detect the increase/decrease of phosphorylation FTZ-F1 in different strains or treatments. Phos-tag SDS-PAGE is an electrophoresis technique capable of separating phosphorylated and non-phosphorylated proteins, it is able to determine the level of phosphorylation and can be carried out easily and at low cost. In this system, divalent metal ions (Mn^{2+} or Zn^{2+}) are added to slow phosphorylated proteins during migration, the higher the amount of phosphorylation, the slower the migration. Western blots can be employed after electrophoresis and can specifically recognize phosphorylated forms of the protein without the need for specific antibodies.

Why the authors did not include the b-actin as a control in fig 6b? The authors claim that “The level of phosphorylated FTZ-F1 was found to be significantly higher in all four resistant strains compared to the susceptible DBM1Ac-S strain” Any statistics analysis has been done to state that? Or maybe it is only a qualitative observation. Again, Why the b-actin was not used as a control in this experiment?

We realize now just how confusing Fig. 6a and Fig. 6b were and appreciate the feedback. In fact, the β -actin control was the same for 6a and 6b but only shown for 6a. We have recombined the figures (now Fig. 5a and 5b), and we hope that this is clear now. The results obtained for the level of expression of FTZ-F1 were only meant to be indicative and not enough biological repeats were performed to provide a statistical analysis.

I totally agree that the silencing of FTZ-F1 in larvae of the resistant strain NIL-R was accompanied by a reduction in both phosphorylated and non-phosphorylated FTZ-F1. But I do not agree with your last conclusion: “These data support the conclusion that the differential phosphorylation of FTZ-F1 in vivo is associated with CryI_{Ac} resistance in P. xylostella”. From my point of view, the results presented in this section suggest that lower levels of FTZ-F1 were accompanied by a decrease in susceptibility CryI_{Ac} but you cannot conclude anything about the role of the differential phosphorylation.

The reviewer is correct that our statement was not justified. We have deleted it from the manuscript. The RNAi experiment on NIL-R does not contradict the conclusion

but certainly does not support it.

- Section: “The MAPK cascade regulates the phosphorylation level of FTZ-F1”
Did you test the susceptibility of the larvae silenced of MAP4K? it would be nice to know the effect of silencing the map4k4 gene, to know if your predictions are correct or not.

We have previously demonstrated the effect of *MAP4K4* silencing on susceptibility but have repeated the experiment for this manuscript (Fig. 6c). The results are the same as found previously and confirm that silencing *MAP4K4* increases susceptibility.

o Fig 7 pane c and f, Are these values significant?

As with the experiment previously described in the original Fig. 6a and 6b (now Fig. 5a, b), these experiments were intended to be just qualitative and indicate whether or not the FTZ-F1 phosphorylation hypothesis tied in with our previous findings on the role of upstream MAPK cascade. Since the data did not contradict this hypothesis (and were not crucial to it), we did not feel the need to perform the necessary repeats to provide statistical significance.

- Section: *Non-receptor genes are conducive to fitness in resistant P. xylostella*
o I will suggest moving lines 292 to 296 to the discussion section.

This results section (Lines 305-311) has been re-ordered to hopefully improve clarity.

o Lines 296-299: “Ectopic expression and cytotoxicity experiments presented here show that Sf9 cells expressing ABCC2 or ABCC3 (but not ABCC1) can bind and be susceptible to CryIAc toxin (Supplementary Fig. 8a, b). These results confirm the distinction between functional receptors and their non-receptor paralogs”. From my point of view, this information is not needed here. I would say that is distracting from the important information.

As reviewer 3 pointed out, the system that we are investigating is a complex, but very elegant, process that can be difficult to understand how it evolved. As we continue to investigate this system, we have performed additional experiments to fill in gaps in the story. Although this manuscript is primarily about the coordinating role of FTZ-F1, we also took the opportunity to test the previously untested hypothesis that ABCC1 is a non-receptor paralog that can compensate for the loss of ABCC2/3 expression upon Bt exposure or in the resistant strains. This was done both by looking at the effects of an ABCC1 knockout and also at the receptor status of this protein. We believe that it makes sense to include both experimental approaches in this section of the manuscript which for the first time demonstrates the role of the non-receptor paralogs in fitness cost compensation – a crucial aspect of the response mechanism. We have though

re-ordered the section to improve clarity (Lines 305-311, Lines 320-322).

I think that results from this section (lanes 300 to 309) should be more emphasized as they are really relevant for this study. As I commented before, in this section there is too much info that can be moved or deleted because it is distracting from the real topic.

Based on this comment and comments from other reviewers, we have rewritten sections of the discussion to hopefully better emphasize the primary findings / significance of our work (Lines 438-447).

We have carefully revised the manuscript according to the reviewers' suggestions and the journal's format. Hopefully, our efforts and the revised manuscript can ease reviewers' concerns and meet the journal's standards. Please contact me if further modifications are required.

Sincerely Yours

Youjun Zhang

Reviewers' Comments:

Reviewer #1:

Remarks to the Author:

The authors have appropriately addressed all of my comments.

Reviewer #3:

Remarks to the Author:

The authors reply all the reviewers' questions very well, I have no other questions.

Reviewer #4:

Remarks to the Author:

After reading carefully the answers to the comments and the changes done in the manuscript by the authors, I think that the manuscript is suitable for publication.

Response to Reviewers' comments

Pilar Morera Margarit
Associate Editor
Nature Communications

September 15, 2022

Dear Editor,

We would like to thank you and all the expert reviewers, for the time spent reviewing our revised manuscript (Manuscript Number: NCOMMS-22-19835A) entitled "A single transcription factor facilitates an insect host combating *Bacillus thuringiensis* infection while maintaining fitness" submitted to *Nature Communications*. We have revised the format of manuscript as required and have highlighted them in yellow. We hope you will find the new version is suitable for publication.

The following is a point-by-point response to the reviewer's comments:

Responses to Reviewers' Comments

Reviewer #1 (Remarks to the Author):

The authors have appropriately addressed all of my comments.

We appreciate the reviewer's comments to help us improve the quality of our manuscript.

Reviewer #3 (Remarks to the Author):

The authors reply all the reviewers' questions very well, I have no other questions.

We appreciate the reviewer's comments to help us improve the quality of our manuscript.

Reviewer #4 (Remarks to the Author):

After reading carefully the answers to the comments and the changes done in the manuscript by the authors, I think that the manuscript is suitable for publication.

We appreciate the reviewer's comments to help us improve the quality of our manuscript.

We have carefully revised the manuscript according to the editor's and reviewers' suggestions and the journal's format requirements. Hopefully, our efforts and the revised manuscript can meet the journal's standards. Please contact me if further

modifications are required.

Sincerely Yours

Youjun Zhang
